# Ocean warming drives rapid dynamic activation of marine-terminating glacier on the west Antarctic Peninsula

Benjamin J. Wallis [1] ✉, Anna E. Hogg [1], Michael P. Meredith [2], Romilly Close[3], Dominic Hardy[3], Malcolm McMillan [3], Jan Wuite [4], Thomas Nagler [4] & Carlos Moffat[5]

Ice dynamic change is the primary cause of mass loss from the Antarctic Ice Sheet, thus it is important to understand the processes driving ice-ocean interactions and the timescale on which major change can occur. Here we use satellite observations to measure a rapid increase in speed and collapse of the ice shelf fronting Cadman Glacier in the absence of surface meltwater ponding. Between November 2018 and December 2019 ice speed increased by $94 \pm 4\%$ $(1.47 \pm 0.6$ km/yr), ice discharge increased by $0.52 \pm 0.21$ Gt/yr, and the calving front retreated by 8 km with dynamic thinning on grounded ice of $20.1 \pm 2.6$ m/ yr. This change was concurrent with a positive temperature anomaly in the upper ocean, where a 400 m deep channel allowed warm water to reach Cadman Glacier driving the dynamic activation, while neighbouring Funk and Lever Glaciers were protected by bathymetric sills across their fjords. Our results show that forcing by warm ocean water can cause the rapid onset of dynamic imbalance and increased ice discharge from glaciers on the Antarctic Peninsula, highlighting the region's sensitivity to future climate variability.

Research on the Antarctic Peninsula (AP) has shown major changes in the region's glaciers and ice shelves throughout the 20th and 21st centuries. The AP has been particularly exposed to the impact of climate change, with near surface atmospheric temperatures on the northern Peninsula warming by +0.54 °C per decade from 1951 to 2011[1]. Similarly, surface waters in the Southern Ocean and Bellingshausen Sea on the west coast of the AP warmed by more than 1 °C from 1955 to 1994[2]. Calving front position measurements show that ice shelves on the AP lost an area of 28,000 km[2] from 1947 to 2008, with major retreat starting in the 1970s and 1980 s[3]. More recent records show that this trend has accelerated, with ice shelves losing an area of 20,500 km[2] between 1997 and 2021[4], with 90% of the Peninsula's 860 tidewater glaciers retreating in the period up to 2010[5]. The associated loss of buttressing from ice shelves and retreat of glaciers has contributed to a substantial increase in ice mass loss from the Antarctic

Peninsula Ice Sheet (APIS)[6–9]. Satellite observations show that ice loss from the region increased by 400 % between the periods 1992 to 1997 $(7 \pm 13$ Gt/yr) and 2007 to 2012 $(35 \pm 17$ Gt/yr), peaking in the decade after the partial collapse of the Larsen-B ice shelf in 2002[10]. Overall, an average of $20 \pm 15$ Gt/yr of ice mass was lost from the APIS in the 25-years between 1992 and 2017[10], and ice loss is projected to increase in the future with 7 to 16 mm of sea-level equivalent by 2100 and 10 to 25 mm by 2200[11].

Both atmospheric and oceanic forcing have driven change on the AP. In 1996, the retreat and collapse of ice shelves was linked to atmospheric warming, specifically the southerly migration of the −5 °C isotherm[12]. Since then, studies have attributed the further retreat and collapse of the Larsen-A and -B Ice Shelves on the eastern AP to thinning, foehn winds and increased summertime melt that led to hydrofracture of crevasses[13–20]. On the west AP, a pattern of increased calving

[1]School of Earth and Environment, University of Leeds, Leeds, UK. [2]British Antarctic Survey, Cambridge, UK. [3]UK Centre for Polar Observation & Modelling, Centre of Excellence in Environmental Data Science, Lancaster Environment Centre, Lancaster University, Lancaster, UK. [4]ENVEO IT GmbH, Innsbruck, Austria. [5]School of Marine Science and Policy, University of Delaware, Newark, DE, USA. ✉e-mail: eebjwa@leeds.ac.uk

front retreat on glaciers in the south has been attributed to the influence of different water masses, with cooler Bransfield Strait Water in the north and warmer Circumpolar Deep Water (CDW) in the south which is transported by the Antarctic Circumpolar Current and intrudes onto the continental shelf from the open ocean[21]. High temporal resolution ice speed measurements from synthetic aperture radar (SAR) satellite imagery have shown pronounced seasonal fluctuations in ice speed of up to 22.3% on west AP tidewater glaciers[22] and glaciers feeding into the George VI Ice Shelf[23]. On the west AP, these speed fluctuations are linked with seasonal ocean warming and surface melt[22,23], showing the sensitivity of this region to short-term variability on sub-annual timescales.

The east and west sides of the AP have significantly different atmospheric and oceanographic conditions. The west AP has high accumulation due to orographically driven precipitation and strong westerly winds[24,25]. There is significant surface melt in the summertime, but less runoff compared with the east AP due to a thick, porous firn layer which takes time to saturate, and which can support perennial firn aquifers[26–28]. South of the Bransfield Strait, CDW flows onto and across the continental shelf, becoming modified CDW (mCDW) through mixing and vertical heat loss to the overlying Antarctic Surface Water layer[29,30]. This contrasts with the much colder waters of the Bransfield Strait and Weddell Sea on the east of the Peninsula[31]. Sea-ice on the west AP follows a seasonal pattern of summertime retreat and wintertime advance, with the wintertime sea-ice cover decreasing strongly in duration since the 1980s[32,33]. In the western Weddell Sea, a

large area of sea-ice regularly persists during austral summer, becoming thicker multi-year ice. Between 1978 and 2015 sea-ice extent (SIE) in the Weddell Sea grew in extent with a small positive trend superimposed upon large interannual variability[34,35]. However, from 2016 onwards this trend reversed, with satellite data showing a reduction in SIE, reaching a record low in 2019 and contributing to record low total Antarctic SIE in 2022[34,36,37].

The future evolution of glaciers on the AP has great significance for broad-scale oceanographic and glaciological changes. Model results show that increased freshwater input to the ocean from precipitation, snow, and glacier melt on the AP can strengthen the Antarctic Coastal Current, leading to enhanced basal melt rates beneath ice shelves in West Antarctica, including Pine Island and Thwaites[38]. The waters of the west AP shelf support highly productive and biodiverse ecosystems which are impacted by changes in sea-ice and meltwater flux[39,40], and glacier retreat exposes new seabed for colonisation by benthic species and enhanced carbon sequestration[41]. The combination of substantial long-term glaciological change, short-term seasonal variability in ice dynamics, and atmospheric and ocean forcing, have a broad impact on Antarctica's ice and oceans and makes the ongoing study and monitoring of the AP a priority.

Cadman Glacier is located on the west coast of the AP south of Anvers Island at 65.6 °S and flows north-west into Beascochea Bay (Fig. 1a). The glacier drains an area of 322 km² from the Bruce Plateau at 2080 m elevation through a steep sided valley, which is 2 km wide at its narrowest point[5,42]. The Cadman Glacier system has four major zones

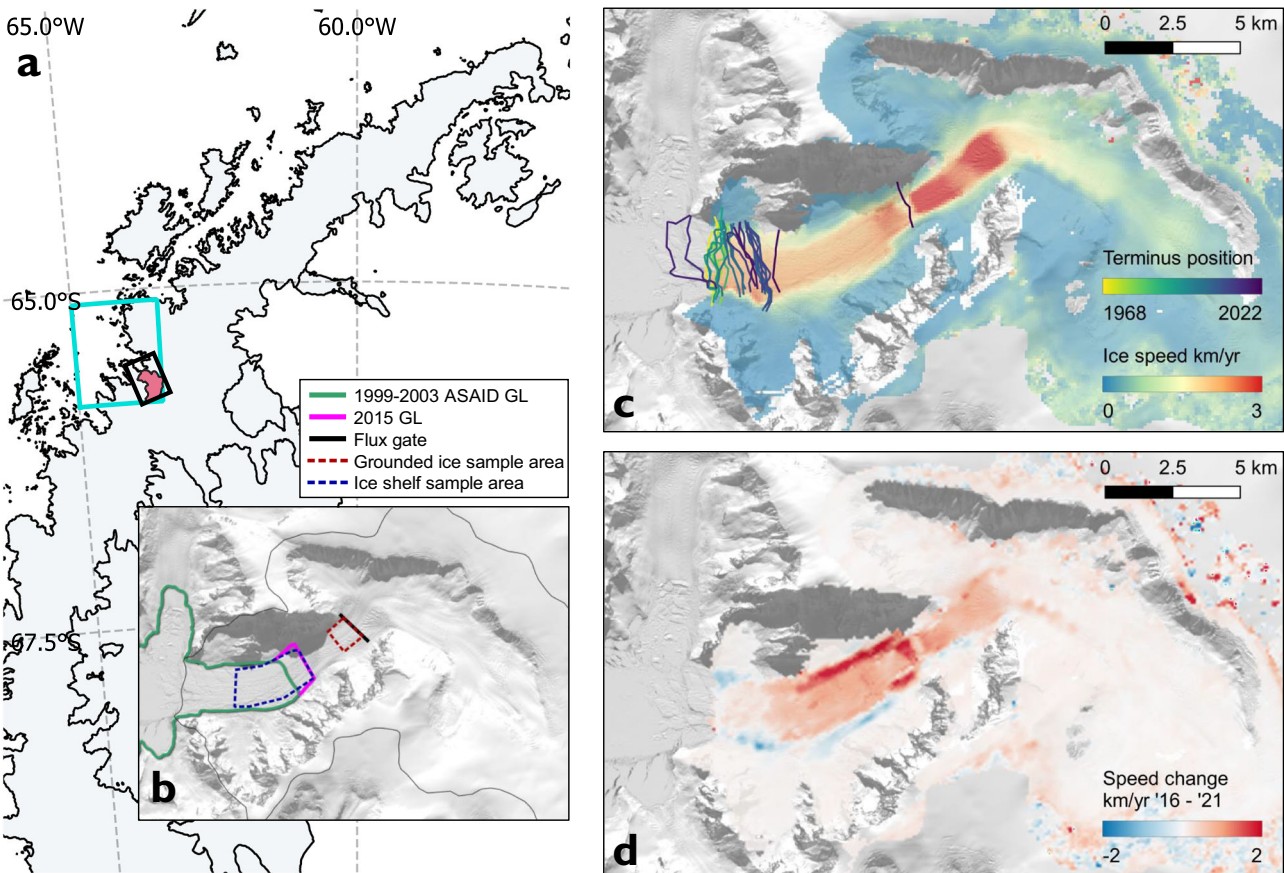

**Fig. 1 | Map of the Cadman Glacier study site showing ice speed and ice speed change. a** Location of Cadman Glacier drainage basin on the Antarctic Peninsula (red shading)[5,87]. The boundary of the ocean reanalysis data sampling region is shown in cyan. **b** Glacier sample areas, including the Cadman Glacier drainage basin (thin black line)[5], Antarctic Surface Accumulation and Ice Discharge grounding line (ASAID GL) (green line)[43], 2015 grounding line (2015 GL) derived from the Reference Elevation Model of Antarctica digital elevation model (REMA DEM)[64] (pink line), ice shelf sampling region (dashed blue line), lower glacier grounded ice sampling region (dashed red line), and the ice discharge fluxgate (thick black line). Background: grayscale Landsat-8, 30th December 2016 (also **c**, **d**). **c** Cadman Glacier ice speed mean (km/yr) 2015–2017 and annual calving front position. **d** Cadman Glacier ice speed change (km/yr) calculated between the 2016 and 2021 annual means.

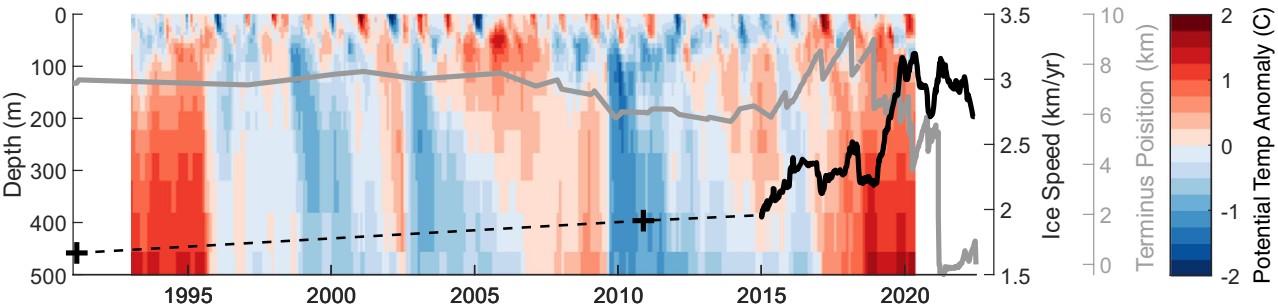

**Fig. 2 | Time-series of ocean potential temperature anomaly, calving front position and ice speed for Cadman Glacier.** Ocean reanalysis potential temperature anomaly from 0 to 500 m depth in Beascochea Bay sample region (Fig. 1a), with time-series of the Cadman calving front location with respect to the most recent measurement (grey line) and ice speed (black line, dashed between observations) also shown. Calving front location is measured using optical satellite imagery from Landsat-5, −7 and −8 for the period up to 2014, then exclusively Sentinel-1 from December 2014 onwards. Ice speed measurements are from Landsat-5 (1991), TerraSAR-X (2010) and Sentinel-1 (December 2014–June 2022) and are extracted in the lower glacier grounded ice sample area (Fig. 1b).

inland from the fjord entrance: i) the Cadman Ice Shelf, ii) the lower glacier which we define as the fastest flowing grounded ice up to 300 m elevation, iii) the Cadman Icefall which is the steepest part of the glacier between 300 and 500 m elevation, and iv) the upper glacier which includes slower flowing ice above 500 m elevation. Satellite image records show that the calving front of Cadman Glacier was remarkably stable throughout the second half of the 20th century, with the ice-front deviating by less than 1 km from a position at the end of its fjord throughout the observational record[5,21]. The glacier catchment has a high rate of surface mass accumulation, up to 4000 mm water equivalent per year, with substantial snowmelt and runoff in the austral summer due to its location on the west coast of the AP[22,24,27]. Due to the snow surface conditions and the speed of ice flow, it is challenging to measure the grounding line location on this glacier using differential interferometry for which coherent Synthetic Aperture Radar (SAR) image pairs are required. Instead, the grounding line position has been measured using altimetry and photogrammetric methods[43]. These observations show that between 1999 and 2003 Cadman Glacier had an ice shelf confined to the lower fjord, where the grounding line was located 6 km inland of the mouth of the fjord.

In this study we measure ice dynamic change on Cadman Glacier over a 31-year period between 1991 and 2022. We use satellite data to measure the calving front position, ice velocity and surface elevation change, which showed that between 2018 and 2019 the glacier accelerated by 94 ± 4 % (1.47 ± 0.6 km/yr), with over 8 km of calving front retreat observed by 2021. We use direct oceanographic measurements and ocean reanalysis data to determine the environmental drivers of this major change, and contextualise it alongside the longer-term evolution of the AP.

## Results
### Change in calving front position
The earliest recorded position of the Cadman Glacier calving front is 1968[44], when the glacier terminated at the entrance to the Cadman Fjord. Prior to 2006, the calving front position did not change significantly, with variability limited to 500 m of retreat from 1985 to 1997, before it re-advanced to the 1984 position by 2001. From 2006 onwards, the glacier underwent a significant retreat of the calving front, moving inland from the fjord entrance by 1.8 km between 2005 and 2009. The glacier calving front remained in this retreated position from 2010 to 2014 (Figs. 1c, 2).

Between 2014 and 2022, 6 to 12-day repeat Sentinel-1 SAR imagery resolves change in the glacier's calving front position with much greater temporal detail than the period from 1968 to 2014. These observations show that the glacier readvanced from May 2015 to February 2018, when it reached its maximum observed extent relative to the full observational record from 1968, extending 700 m beyond

the mouth of its fjord into Beascochea Bay. Although advanced, during this period the seaward 3 km of glacier tongue shows a substantial increase in damage and crevassing on the ice shelf surface and a widening of the shear margins (Fig. S1), suggesting that the structural integrity of the ice was decreasing. Notably in the 17th January 2020 image (Fig. S1d) there is a distinctive along-flow damage feature on the central ice shelf, which may indicate channelised sub-shelf melting. The glacier calved and retreated by 2.8 km coincident with the start of its period of most rapid acceleration which was sustained throughout 2019. From March 2019 to March 2021, the glacier underwent a cycle of calving and readvance in this retreated position, while flowing approximately 1 km/yr (44 %) faster than pre-2019. Throughout this period of faster flow, the floating ice shelf became increasingly damaged and crevassed, until eventually breaking up into large tabular blocks. Notably, although summer melt is widespread in the region, we observe no surface meltwater ponds visible on the ice surface in any optical satellite image for our period of study (1984–2021), which we attribute to the north-west AP's thick firn layer and high firn air content[45]. Cadman Ice Shelf completely collapsed in March 2021, with the calving front location retreating by a further 5.2 km to a position 2.4 km inland of the ASAID (1999–2003) grounding line location, and it remained in this position for the remainder of the study period to July 2022.

### Ice velocity from 1991 to 2022
Our ice velocity measurements show the rate of ice flow on Cadman Glacier at high spatial and, since 2015, temporal resolution (Fig. 1b). The grounded ice flows at speeds above 1 km/yr 10 km inland of the 1999–2003 grounding line position and the highest speeds of more than 4 km/yr are observed on the Cadman Glacier icefall and lower glacier in 2021/22. This makes Cadman Glacier the fastest flowing glacier on the west AP, where average flow speeds are approximately 1 km/yr, and comparable to the largest glaciers on the AP such as Fleming Glacier which flows at 2.5 km/yr[22].

Our 30-year timeseries of ice velocity (Figs. 2, 3a, sample areas defined Figs. 1b, 3b) on Cadman Glacier shows substantial changes in flow speed throughout the study period. The grounded glacier accelerated from 1.67 ± 0.27 km/yr in February 1991 to a peak of 3.20 ± 0.05 km/yr in May 2022, a speed increase of 91 ± 17 % (1.53 ± 0.29 km/yr). The floating ice shelf experienced a greater acceleration, from 1.33 ± 0.24 km/yr in 1991 to 3.87 ± 0.05 km/yr in May 2022, an acceleration of 191 ± 19 % (2.54 ± 0.25 km/yr). The majority of this speed increase (1.19 km/yr on the grounded ice and 2.01 km/yr on the shelf) occurred in the period from December 2014 to June 2022 (Fig. 1d). This included a major acceleration event from November 2018 to December 2019 where flow speeds increased by 44 ± 3 % (0.96 ± 0.6 km/yr) on the grounded glacier and 94 ± 4%

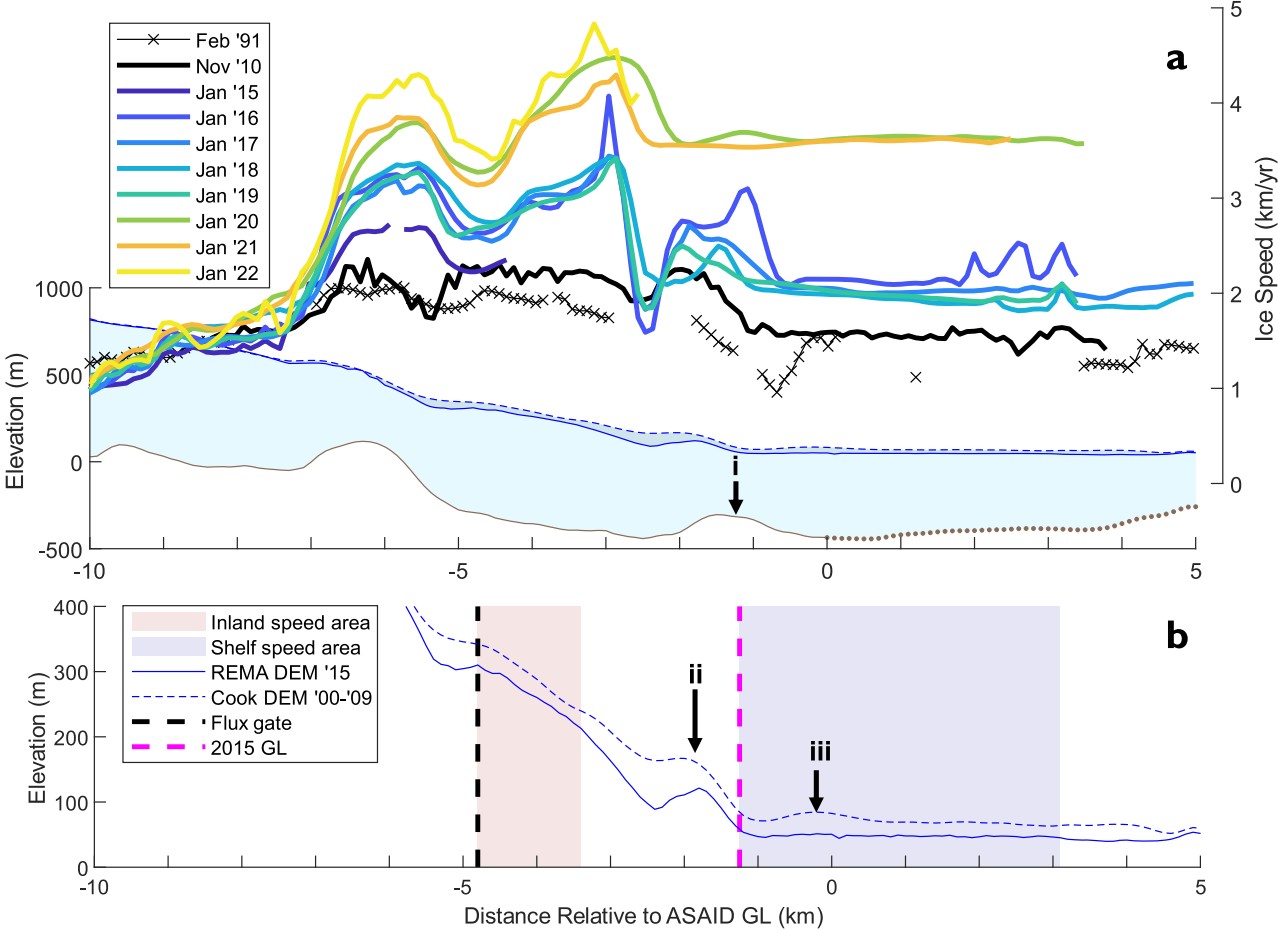

**Fig. 3 | Cadman Glacier ice speed and geometry in the flowline direction.**
**a** Flowline profile of Cadman Glacier showing change in ice speed from 1991 (black crosses) to 2022 (yellow line), with horizontal distance on the x-axis given with respect to the Antarctic Surface Accumulation and Ice Discharge (ASAID) (1999–2003) grounding line location (illustrated in Fig. 1b)[43]. Ice speed measurements are shown for February 1991 (Landsat-5), November 2010 (TerraSAR-X), and January 2015 to January 2022 (Sentinel-1). Glacier geometry is shown using measurements of the ice surface elevation from the Reference Elevation Model of Antarctica digital elevation model (REMA DEM)[64] (solid blue line) and the Cook

Antarctic Peninsula digital elevation model (AP DEM)[83] (dashed blue line), with bed elevation (grey line) shown from Huss and Farinotti[72]. Point (i) is the 2015 bed ridge. **b** High vertical-resolution surface elevation geometry shown to highlight the point (ii) 2015 surface bump and point (iii) the 1990s surface bump. Surface elevation is again shown using the REMA DEM (solid blue line) and Cook AP DEM (dashed blue line), with the lower glacier sampling region (red shading), ice shelf sampling region (blue shading), the ice discharge fluxgate position (black dashed line), 2015 grounding line (pink dashed line) also annotated (Fig. 1b).

(1.47 ± 0.6 km/yr) on the floating ice shelf. In contrast, the neighbouring glaciers in Beascochea Bay, Lever and Funk Glaciers, have not accelerated over the same time period (2014 to 2021). Instead, they have maintained a constant ice speed with small seasonal fluctuations. Lever Glacier flowed at a mean speed 0.84 ± 0.04 km/yr and Funk at 1.22 ± 0.03 km/yr (Fig. S2), suggesting that the drivers of Cadman Glacier's 2018/19 acceleration are localised and specific to this glacier.

In addition to this overall acceleration, the spatial pattern of ice velocity on Cadman Glacier changed during the study period. In 1991, there was a local speed minimum of 0.96 km/yr in the lower glacier section 700 m inland of the grounding line location (Fig. 3a). From 2010 to 2022, this local speed minimum is absent, with ice speed constant throughout this section instead. By 2010, another local speed minimum of 1.91 km/yr is observed 2.5 km upstream of the 1999–2003 grounding line. Although it was not possible to measure ice speed on this section of the glacier in 1991, this local speed minimum is also observed in December 2014 at the start of our Sentinel-1 velocity timeseries. This ice speed minimum persists until the glacier's rapid acceleration in 2018/2019, when by mid-2019 it has transitioned from a distinct local minimum to a zone where ice speed transitions to flow at a constant 3.5 km/yr, which we interpret as being indicative of the

downstream floating ice shelf. In March 2021, the Cadman Ice Shelf completely collapsed, with the glacier calving front retreating to a position 2.4 km inland of the 1999–2003 grounding line. Over the 30-year study period, the observed change in ice dynamics indicates that Cadman Glacier's grounding zone configuration has undergone substantial change. Specifically, the loss of the most seaward local ice speed minima between 1991 to 2010 and 2018 to 2019 (Fig. 3a) suggests that there has been a loss of buttressing resistive force which would have been local to both these speed features.

## Surface elevation change

Surface elevation measurements from CryoSat-2 were used to assess thinning on the Cadman Ice Shelf from 21st November 2010 to 10th March 2019. Between November 2010 and March 2019 prior to the ice speed acceleration, we observed a total of −16.2 ± 4.3 m surface elevation change in a sample region covering the floating ice shelf (Figs. 1b, 4a), an average annual rate of −1.95 ± 0.53 m/yr (Fig. 4b). Assuming the ice shelf is in hydrostatic equilibrium, has an average ice density 917 kg/m³, and that mass losses are entirely driven by basal melting rather than changes in the firn layer, implies a total ice shelf thinning rate of 18.5 ± 5.0 m/yr[46,47]. This result compares well with a

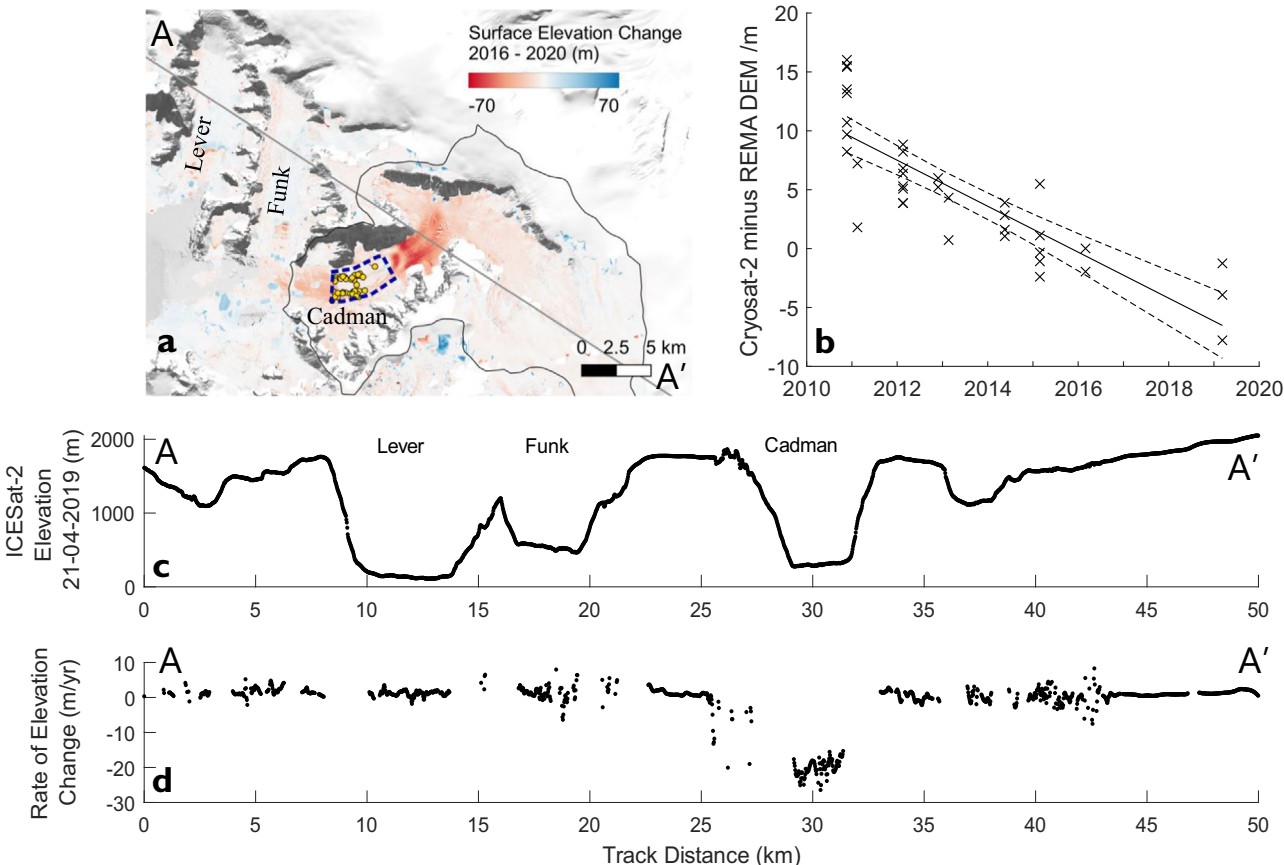

**Fig. 4 | Surface elevation change on Cadman Glacier from repeat REMA DEMs, CryoSat-2 and ICESat-2. a** Map of Cadman Glacier with surface elevation change data from the Reference Elevation Model of Antarctica digital elevation model (REMA DEM) tiles acquired between 2016-11-10 and 2020-02-25. ICESat-2 ground track (grey line) (Fig. 4c, d), ice shelf region where CryoSat-2 elevation change retrievals were extracted (blue dashed box), and location of CryoSat-2 Point of Closest Approach (POCA) elevation measurements (gold dots) are also shown. Base image is Landsat-8 30th December 2016. **b** Timeseries of elevation change from 2010 to 2021 calculated using CryoSat-2 elevation measurements differenced against the REMA DEM[64] (black crosses), with a linear fit (solid black line) and 95% confidence interval (dashed black lines) also shown. **c** Elevation profile for ICESat-2 track A-A' acquired on 21st April 2019, with the locations of Lever, Funk and Cadman Glaciers also shown. **d** Rate of surface elevation change along ICESat-2 track A-A' measured between 21st April 2019 and 17th August 2021.

previous study that estimated surface lowering at a rate of 2.23 m/yr between 2001 and 2010 using laser altimetry and stereo image DEMs, albeit this study included all floating and grounded ice below 1 km elevation[48]. Nonetheless, these results show a consistent trend in long-term thinning of Cadman Glacier's floating shelf from 2001 until its collapse in 2021.

Comparison of two co-registered, but temporally separated DEMs (Cook AP DEM 2000–2009 and REMA DEM 2015) shows the spatial pattern of surface elevation change on Cadman Glacier prior to the 2018/19 ice speed acceleration event. The elevation profile (Fig. 3b) shows that a surface bump is visible in the Cook DEM 500 m inland of the grounding line location. This bump (referred to as point (iii) in Fig. 3b, henceforth 1990s bump), located 2 km inland of the ASAID grounding line, is coincident with a local ice speed minimum and region of compressive stress in the 1991 ice speed profile. This can be interpreted as either indicative of the slope boundary layer, where the transition from vertical shear stress dominated flow to extensional stress dominated flow creates the distinctive grounding zone bump through viscous beam behaviour, or more simply grounding on a bed slope ridge[49]. By 2010, the local ice speed minimum was replaced by a region of constant ice speed and the 2015 REMA DEM shows that the point (iii) surface bump has disappeared. Additionally, taking the break in surface slope in the REMA DEM as an approximation for the location of Cadman Glacier's grounding line in 2015, we find that in 2015 the grounding line is 1.25 km inland of the ASAID 1999-2003 position

(Fig. 3b). These changes between the Cook DEM (2000-2009) and the REMA DEM (2015) further support our interpretation from ice velocity measurements that the grounding line of the glacier retreated between 1991 and 2010.

Further inland, the DEMs show another surface slope inflection point 2.5 km up-glacier of the ASAID grounding line, called point (ii) (Fig. 3b, henceforth 2015 bump). This inflection point deepens significantly in the 6-year period between the two DEMs (approximately 2009 to 2015). Again, we interpret this to be a feature of the slope boundary layer or a surface manifestation of a bed ridge. The presence of the point (ii) surface bump in 2015 following the point (iii) bump present in the 1990s, and given that the 2015 bump is several coupling lengths up-glacier of the 1999–2003 grounding line position, we favour the interpretation that this feature is the surface manifestation of a bed ridge. This corresponds to a bed ridge, point (i), at the same location in the modelled bed topography (Fig. 3a) and is supported by our ice velocity measurements which show an ice speed minimum immediately inland of this surface bump from 2010 to 2018.

Since Cadman Glacier's rapid acceleration in 2018/19, there have also been substantial changes in the surface elevation on the inland grounded section of the glacier. Repeat track laser altimetry data from ICESat-2 between 21st April 2019 and 17th August 2021, acquired in the period immediately following the 2018/19 ice speed acceleration event show a maximum surface elevation change rate of −26.4 ± 5.1 m/yr at the base of the Cadman Icefall (Fig. 4d), and a mean rate of

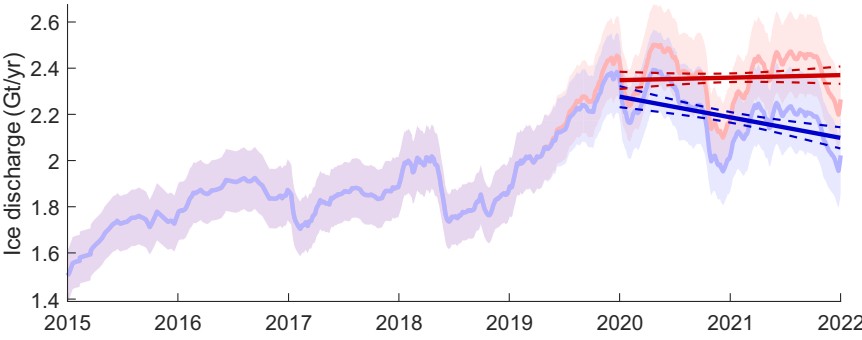

**Fig. 5 | Rate of ice discharge from Cadman Glacier from 2015 to 2022.** Time-series of ice discharge measured using time variable ice speed measurements along the flux gate on Cadman Glacier (Fig. 1b). Discharge is calculated both with the ice surface elevation lowering rate of 20.1 m/yr applied from 21st April 2019 (blue line) and without surface lowering term (red line). Shading shows the errors for both discharge time-series. The linear fit to each time-series is also shown from 2020 onwards (dark red & blue line) with the 95 % confidence intervals also indicated (dashed lines).

−20.1 ± 2.6 m/yr across the 2.2 km glacier width. The mean ice thickness of this transect is 313 m, which shows that the mean percentage thinning rate is 6.4%/yr, with the maximum percentage thickness change rate of 8.6%/yr observed in the glacier margins where ice is thinnest.

To determine the spatial extent of glacier thinning, we complemented our ICESat-2 measurements with estimates of surface elevation change derived from repeat 2 meter resolution REMA Digital Elevation Models that were acquired in November 2016 and February 2020[50]. These measurements show that the main trunk of the glacier thinned by a maximum of 67.1 ± 1.5 m during this 3.3-year period, with the majority of ice volume loss occurring within a region extending 8 km upstream of the grounding line (Fig. 4a). Over the lower glacier grounded ice sampling region (Fig. 1b) the mean rate of surface elevation change was −12.8 ± 0.5 m/yr, with a maximum rate of −17.1 ± 0.5 m/yr. These data also support our measurements of the thinning of Cadman Ice Shelf, where between November 2016 and February 2020 we measure a surface elevation change of −1.61 ± 0.46 m/yr, in agreement with the rate of −1.95 ± 0.53 m/yr from 2010 to 2019 measured with CryoSat-2.

Together, these elevation change results show that there has been an exceptionally high rate of ice thinning across Cadman Glacier's drainage basin, in stark contrast to the relative stability and absence of any observed change in ice thickness on neighbouring Funk and Lever Glaciers which also did not change speed during this period (Fig. 4a, d, Fig. S2a, b). The coincident timing of these thinning observations with the ice speed change measurements demonstrates that this is a dynamically induced thinning caused by the 2018/19 ice acceleration, initiated by the loss of basal traction over the 2015 bed ridge.

## Ice discharge

We used the ice speed and surface elevation change observations to calculate change in ice discharge from Cadman Glacier over the 30-year study period. Our results show that overall, ice discharge increased significantly by 49.6%, from 1.34 ± 0.11 Gt/yr in 1991 to 2.00 ± 0.16 Gt/yr in December 2021 (Fig. 5). Following an initial increase from 1991 to 2010 (1.57 ± 0.13 Gt/yr), ice discharge was stable between 2010 and December 2014 as a consequence of there being no speed change and no evidence of ice thickness increase in this period. By January to March 2015, mean ice discharge remained at 1.57 ± 0.11 Gt/yr, despite being a period where seasonal speed fluctuations would be at their maximum. Throughout 2015 ice discharge increased until it reached a mean of 1.85 ± 0.13 Gt/yr between 2016 and 2018 with no positive or negative trend. Ice discharge increased rapidly by 28.1% from December 2019 as ice speeds accelerated, reaching a maximum discharge rate of 2.37 ± 0.17 Gt/yr by January 2020. While ice speed remained high throughout 2020 to 2022, ice discharge

decreased over this period due to the extremely high rate of surface elevation change. By December 2021 ice discharge was 2.00 ± 0.16 Gt/yr, with surface lowering (measured from ICESat-2) accounting for a reduction in the discharge rate of 0.25 ± 0.23 Gt/yr (Fig. 5). The importance of accounting for ice surface lowering term is demonstrated by fitting a linear trend to the post acceleration timeseries of ice discharge for two years from 1st January 2020 to 31st December 2021. When ice thickness change is accounted for, our results show a trend in the ice discharge of −0.09 ± 0.02 Gt/yr² as the glacier evolves toward a state of dynamic balance, in contrast to calculations of discharge without the surface lowering term applied (Fig. 5).

## Ocean Temperature 1993–2021

We use global ocean reanalysis output and direct oceanographic station measurements to investigate the role of thermal forcing from warm ocean water at Cadman Glacier. We extracted ocean temperature data from a region up to 80 km offshore from Cadman Glacier (Fig. 1a), to produce a time-series of ocean potential temperature anomaly from the reanalysis (Fig. 2). In this time-series, the top 50 m of the water column exhibits seasonal ocean temperature fluctuations driven by atmospheric conditions. In contrast, longer term fluctuations in temperature are apparent below this depth, with positive potential temperature anomalies of up to 2 °C present between 1993–95, 2004–2009 and 2016–2021. This follows an approximately cyclical pattern on decadal timescales, with the strongest anomalies in 1993–95 and 2016–2021. We see significant warm temperature anomalies from 50 to 500 m in the water column from 2017 onwards, which coincide with the period of ice shelf retreat and ice speed acceleration observed on Cadman Glacier. In-situ observations from the Palmer Long-Term Ecological Research (PAL-LTER) programme show that the highest January temperatures for the 27-year period between 1993 and 2020 occurred in 2019 for the Anvers Hugo Trough (AHT) and Palmer Deep (PD). The observations also show a significant positive salinity anomaly in the AHT in 2019 (Fig. 6c, d), which indicates a shoaling of the pycnocline in January 2019, where warm CDW was able to flow over a topographic boundary into PD and likely further onwards to Cadman Glacier. Without direct measurements in Beascochea Bay, it is difficult from these data alone to ascertain unambiguously the duration of this shoaling or the temperature anomaly of the water that impinged ultimately on Cadman Glacier. However, they demonstrate that deep water conditions in the shelf region upstream of Cadman Glacier exhibited variability with the timing and magnitude consistent with strong oceanographic forcing of the glacier system.

The pathway for warm ocean water to circulate across the west AP shelf and reach Cadman Glacier is complex. Bathymetric data show that Beascochea Bay is up to 760 m deep, with a historic grounding zone wedge with a minimum depth of 360 m crossing the bay 7 km

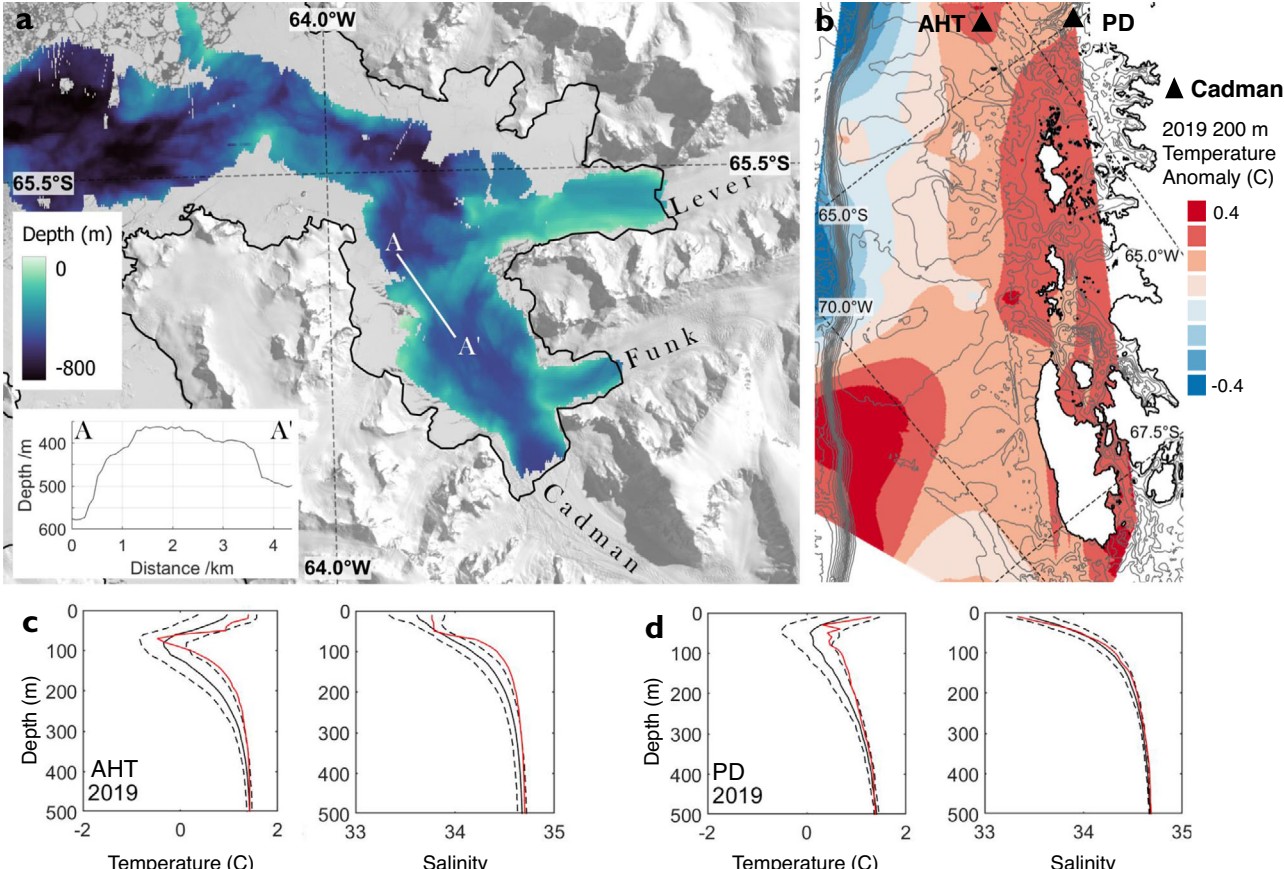

**Fig. 6 | Beascochea Bay bathymetry and west Antarctic Peninsula shelf ocean temperature change. a** Bathymetry of Beascochea Bay, showing the mid bay ridge (A-A'), A-A' depth profile insert. Base image is Landsat-8 30th December 2016. **b** Ocean temperature anomaly vs the 1993 to 2010 mean at 200 m depth from gridded Palmer Long-Term Ecological Research programme station data. The location of Cadman Glacier is marked, alongside the locations of the Anvers-Hugo Trough (AHT) and Palmer Deep (PD). 250 m depth contours from (International Bathymetric Chart of the Southern Ocean (IBSCO) v2[95]. **c** Ocean temperature and salinity (practical salinity units) profiles at the AHT station (64.59 S, 65.35 W), 1993–2020 January mean and standard deviation (black and dashed) and 2019 profile (red line). **d** Ocean temperature and salinity (practical salinity units) profiles at the PD station (64.95 S, 64.27 W), 1993–2020 January mean and standard deviation (black and dashed) and 2019 profile (red line).

seaward of the Cadman Fjord entrance (Fig. 6a). From the bay to the continental shelf, there is a bathymetric high with a minimum depth of 300–350 m separating the deep water of Beascochea Bay from the west AP shelf. There is only a minor bathymetric sill at the entrance to Cadman Fjord, with a minimum depth of 460 m, therefore, any warm water masses entering Beascochea Bay can continue relatively unimpeded to access Cadman Glacier's shelf cavity and grounding zone. In contrast, neighbouring Funk and Lever Glaciers are shielded from deeper water by seaward bathymetric sills at depths of 230 m and 200 m respectively, meaning they are substantially more protected from incursions of warm ocean water in comparison to Cadman Glacier.

## Discussion

Cadman Glacier is unusual for glaciers on the west AP as it had a significant ice shelf confined within a narrow fjord, which survived into the 21st century. North of Marguerite Bay, only Talbot, Lever, Smith and Lawrie glaciers, and the Muller Ice Shelf have floating tongues or ice shelves longer than 3 km (evaluated using the ASAID grounding line dataset). Cadman Glacier's location at 65.6 °S places it in a region of the AP where warm ocean waters and intrusions of CDW have been linked to glacier retreat[3,21]. In the wider AP region, studies have shown that glaciers and ice shelves began losing area in the 1970s and that this trend continued until at least 2015. Cadman Glacier defied this trend until 2006, remaining in a consistent configuration terminating at the

end of its fjord throughout the 20th century. However, this lack of retreat does not preclude the ice shelf thinning on Cadman Ice Shelf during this period, with our analysis of CryoSat-2 data and previous studies[5,21] showing that sustained thinning occurred between 2001 and 2019 (Fig. 4a, b). This, combined with previous studies in the area, suggests that the glacier experienced persistent ocean driven thinning of its ice shelf from at least the early 2000s, but more likely since the 1970s[3,21]. Our results suggest that this thinning only reached a sufficient cumulative level to make the glacier vulnerable to ice front retreat from 2006 onwards, and we see further evidence of the degradation of the structural integrity of the ice shelf during this period through the change in damage observed on the ice shelf surface (Fig. S1).

Long-term thinning on the Cadman Ice Shelf can cause ice speed up through the reduction in buttressing support, likely caused by ice ungrounding from bed topographic features and a reduction in lateral resistive stresses as ice at the grounding zone thinned to the point of floatation. Although no updated direct measurement grounding line data exists for our study period, our surface elevation, elevation change, and ice velocity observations support the hypothesis that warm ocean temperatures have caused increased melting and thinning of the glacier's ice shelf throughout the 31-year study period. Between 1991 and 2010 there was a change from a local ice speed minimum feature to an area of spatially constant ice speed immediately behind the ASAID grounding line location, which was matched by the loss of

an ice surface bump between the Cook AP (2000–2009) and the REMA DEMs (2015), together with the inland migration of the break in surface slope. We interpret these changes to be indicators of a loss of basal traction and the retreat of the grounding line, from the ASAID grounding line location to a position 1.25 km further inland located near the point (ii) 2015 surface bump and associated speed minimum (Fig. 3). Such a loss of basal traction could also explain the increase in ice speed of 0.25 km/yr (15%) between 1991 and 2010.

Following this long-term thinning and possible grounding line retreat our results show that Cadman Glacier underwent a period of far more rapid dynamic change from 2018 to 2022, where grounded ice accelerated by 0.96 km/yr (44%) from November 2018 to December 2019 and the floating ice shelf disintegrated entirely in March 2021. Surface velocity features again provide an explanation for this acceleration, as a local ice speed minimum which we associate with grounding on a bed ridge, disappeared during the period of maximum acceleration between 2018 and 2019. We interpret this as the loss of basal traction from a bed ridge and therefore conclude that thinning of the Cadman Ice Shelf and melting at the grounding zone caused the glacier to reach floatation in this section and unground from the 2015 ridge (point ii), leading to rapid grounding line retreat across a retrograde bed slope and acceleration due to the loss of basal drag and resistive stress. The Cadman Ice Shelf fully collapsed in March 2021 due to this thinning, possibly intensified through channelised melting, and the loss of pinning on the 2015 ridge. The absence of surface melt ponding on the Cadman Ice Shelf prior to its collapse demonstrates that surface ponding is not a necessary precondition for ice shelf collapse in the Antarctic Peninsula, which can occur through ocean-driven thinning and unpinning alone, similar to small ice shelf failures observed in Greenland[51].

Since Cadman Glacier's 2018/2019 acceleration event, the glacier has been in a state of substantial dynamic imbalance. Our ICESat-2 measurements show that since peak ice speed was reached in 2019, there has been a surface elevation change of $-20.1 \pm 2.6$ m/yr on Cadman's fast flowing lower glacier (Fig. 5a, c). No equivalent change is detected for neighbouring Funk and Lever Glaciers which are covered by the same ICESat-2 track, indicating that this is ice dynamic thinning caused by Cadman Glacier's increased flow speed. The dynamic thinning has propagated inland beyond the Cadman Icefall to the upper glacier, where a surface lowering rate of $3.0 \pm 0.5$ m/yr is observed 8 km inland of the grounding line. Without a significant change in environmental forcing that would enable the glacier to readvance, the current mass imbalance is likely to continue until the glacier reaches a new equilibrium through a combination of dynamic thinning, changes in its driving stress, and reduced accumulation due to surface lowering. In the last complete year of measurements (2021), Cadman Glacier's mean discharge rate was $2.16 \pm 0.17$ Gt/y, an increase of $17.9 \pm 12.5\%$ ($0.30 \pm 0.21$ Gt/yr) in comparison to the period immediately before the 2018/19 acceleration event, and an increase of $60.6 \pm 14.7\%$ ($0.81 \pm 0.20$ Gt/yr) compared to 1991. Although ice discharge has decreased from its post-acceleration peak due to dynamic thinning, it is unclear when the glacier reaches a new equilibrium, or even if it can reach a new equilibrium should environmental forcing change faster than the glacier's response time. However, even with ice speed unchanged from 2022 values, at the current rate of dynamic thinning ($-20.1 \pm 2.6$ m/yr across our flux gate) ice discharge will return to its 2016–2019 mean by October 2024.

Our results suggest that the synchronous timing of the warm ocean water forcing can explain the rapid ice dynamic change, ice shelf collapse, and subsequent dynamic imbalance observed on Cadman Glacier. Ocean temperature from reanalysis output shows that from 2013 onwards a positive temperature anomaly was present at all depths accessible to melt the ice shelf and grounding zone (Fig. 2). This oceanic warm phase intensified through 2016 to 2018, as confirmed by direct ocean temperature measurements, providing an explanation for the timing of the 2018/19 acceleration event. In particular, these data show that January 2019 was a period of anomalously high ocean temperature across the shelf (Fig. 6b) and locally in the Anvers-Hugo Trough and Palmer Deep (Fig. 6c, d). Similarly, reanalysis data shows a positive temperature anomaly initially at shallow depths before Cadman Glacier's smaller 2008/9 calving front retreat, later progressing to deeper depths and persisting until 2009. This suggests a coupling between Cadman Glacier's ice dynamics and decadal-scale ocean temperature anomalies on the west AP shelf, similar to connections with large-scale climate variability identified on large ice shelves in West Antarctica[52–54]. This provides evidence that decadal variability in the Southern Ocean influences ice dynamics around the Antarctic coastline as far as the west Antarctic Peninsula, even on smaller tidewater glaciers[54].

The sensitivity of Cadman Glacier, but not neighbouring Funk and Lever Glaciers, to these ocean temperature fluctuations supports our conclusion that basal melting and ungrounding of Cadman Glacier is responsible for the 2018/19 acceleration. The entrances to the Lever and Funk fjords have bathymetric sills at depths of 200 m and 230 m respectively, which limit the access of deep warm water to these glaciers compared with Cadman, so they are likely to be less sensitive to ocean forcing than Cadman Glacier which does not have a sill higher than the 350 m mid bay ridge. Sills are an effective mechanism to restrict deep, warm oceanic waters to marine-terminating glaciers, and modelling work shows that sills deeper than warm water layers can still limit basal melt rates where a hydraulically controlled circulation regime dominates[55,56]. Our results suggest differences in bathymetric structure played a strong role in the different response of Cadman and neighbouring glaciers to warming shelf water, however in future Lever and Funk Glaciers may become substantially more sensitive to ocean forcing if temperature anomalies are greater and they progressively thin in an evolution which is similar to Cadman Glacier's but delayed. This is consistent with studies in Greenland, where bathymetric sills are important controls on ocean-fjord exchange and basal melt rates, such that the divergent behaviour of neighbouring glaciers can be explained by differences in fjord bathymetry[57–59]. Therefore, the future evolution of glaciers on the west AP coastline will likely depend strongly on the bathymetric configuration of individual embayments, adding complexity to the challenge of making accurate predictions of future mass loss.

The rapid dynamic activation of Cadman Glacier in 2018/19 can be understood conceptually as an example of a glaciological tipping point, where an over-deepened bed geometry creates a bifurcation of steady states[60]. Long-term, decadal thinning of the ice shelf and melting at the grounding zone conditioned the susceptibility of the glacier to reach floatation from its bed ridge, placing it close to the bifurcation point between advanced and retreated states. This tipping point was reached in 2018, most likely caused by the kick of an intense episode of warm water incursions onto the west AP shelf which caused the ungrounding, rapid acceleration and retreat. Our results demonstrate that such tipping points can increase ice discharge extremely rapidly, in Cadman Glacier's case by 28.1% in 13 months. Other glaciers on the Antarctic Peninsula may be vulnerable to similar sudden acceleration and retreat, where bed conditions and bathymetry allow, similar to heterogeneous patterns of tidewater glacier retreat controlled by bed topography observed in Greenland[61,62]. For example, the largest glacier on the west coast, Fleming Glacier, has a floating tongue and is grounded on a retrograde bed slope, similar to Cadman Glacier[63]. Despite the relatively small size of the Antarctic Peninsula Ice Sheet compared to the West and East Antarctic Ice Sheets, the stability of the AP's glaciers is important to global sea-level. The region has contributed $1.4 \pm 0.37$ mm to global mean sea-level from 1992 to 2017, 19% of the Antarctic total[10]. Additionally, increased freshwater input from ice discharge and newly exposed seabed due to glacier retreat would have regional-scale consequences for ocean circulation, productivity, carbon sequestration, and the ecosystem[38,40,41].

Overall, the acceleration, retreat and dynamic thinning observed on Cadman Glacier since 2018 shows that glaciers on the Antarctic Peninsula can become imbalanced and increase their ice discharge into the ocean rapidly in response to forcing by warm ocean waters, highlighting the regions sensitivity to future climate variability. Our results reveal the speed and magnitude at which such glacier responses can occur and show that bed geometry and local bathymetric features are important controls on the vulnerability of glaciers in the region to such acceleration and retreat, however these glaciers are often poorly observed in terms of near ice front bathymetry and radar surveys of bed topography.

This study represents a uniquely detailed assembly of data allowing us to observe and characterise the forced transition of an Antarctic glacier from one with a small floating shelf to a tidewater glacier. It highlights the impact of continuous, high-resolution multi-decadal climate observations and remote sensing campaigns which have contributed to our understanding of the complex interaction of Earth-system processes and how Antarctica will respond to a warmer climate. This work also shows the importance of continuing and improving these observations. Dense time-series of ice velocity measurements, such as those from Sentinel-1, could be further enhanced by dedicated campaigns of in-situ validation to quantify the impact of surface processes on velocity measurement error, allowing greater sensitivity to detect short-term ice dynamic changes. Substantial further insights would also be gained from frequent grounding line position measurements, particularly in regions such as the AP where low SAR coherence currently makes these measurements difficult. Improving the temporal and spatial resolution of altimetry measurements will improve the monitoring of ice shelf thinning across the Antarctic continent. Expanded airborne ground penetrating radar campaigns to measure ice thickness and bed topography near the grounding zone would reduce uncertainty in ice discharge and better characterise which glaciers are vulnerable to dynamic tipping points like the one observed in this study. Furthermore, Cadman Glacier and the Beascochea Bay system including Funk and Lever Glaciers are strong candidates for further studies and fieldwork measurements. New and sustained oceanographic measurements of the bay and individual glacier fjords behind the bathymetric sills would be valuable to understand heat transport across the west AP shelf and to compare circulation between Greenlandic fjords and the AP helping to better understand the future evolution of the Antarctic Peninsula's glaciers.

## Methods
### Ice velocity and discharge
We measured ice velocity on Cadman Glacier (Fig. 1a) by using the full time-series of Sentinel-1a and −1b synthetic aperture radar (SAR) Interferometric Wide (IW) swath mode single look complex (SLC) image pairs acquired every 6 to 12-days between December 2014 and July 2022. The images were co-registered using the REMA 200 m Antarctic digital elevation model (DEM)[64] and precise orbit determination (POD) data from the European Space Agency. We tracked the displacement of surface features using the frequency domain intensity cross-correlation technique[65,66] and converted this to ice velocity using the REMA 200 m DEM assuming surface parallel flow. The results were posted on a 100 m grid, and a spatially variable ice velocity error estimate was calculated by multiplying the signal to noise ratio of the cross-correlation with the ice speed[66,67].

To assess change in Cadman Glacier's ice speed (Fig. 1b), we extracted a timeseries of velocity measurements and associated errors along the glacier's central flowline and averaged in the ice shelf and lower glacier grounded ice sample boxes defined in Fig. 1b. This data is post-processed using SNR filtering at a threshold of 5.8[68] and a Bayesian recursive smoother[22] to compute the maximum likelihood estimate for ice speed and associated uncertainty at daily timesteps for the profile. In addition, we processed historical ice speed measurements using the same feature tracking technique applied to a pair of Landsat-

5 images from 9th February 1991 and 25th February 1991, and a triplet of TerraSAR-X images from 10th November 2010 to 2nd December 2010. Optical feature tracking of Landsat images was performed with the ImGRAFT toolbox[69].

We used the ice velocity measurements to calculate ice discharge on Cadman Glacier by integrating ice velocity across a flux gate defined 2.2 km inland from the post-collapse calving front position in January 2022 (Fig. 1b where coverage across all velocity datasets is good and where the ice surface elevation change can be accurately measured using repeat tracks of the ICESat-2 laser altimeter[10,70]. Flow directions were taken from the MEaSUREs ice speed mosaic dataset[71] and ice thickness was calculated from the difference between the bed elevation[72] and the ice surface elevation using the REMA 100 m DEM[64]. We apply a surface height correction to the ice thickness data using the mean thinning rate calculated across the flux gate between 21st April 2019 and 17th August 2021, to account for the observed surface lowering during the study period (Fig. 5). A time varying firn air content correction is also applied to the ice thickness data using the IMAU firn densification model[73].

### Calving front location
We measured the change in calving front location on Cadman Glacier by manually digitizing the calving front position in Landsat-5, Landsat-7, Landsat-8 and Sentinel-2 optical imagery, and Sentinel-1 SAR imagery using the GEEDiT digitization tool[74], where the availability of optical images in limited by cloud cover and a lack of daylight in the Antarctic winter. This produced a time-series of calving front positions at variable intervals between 1984 and 2014, and a continuous time-series from December 2014 to July 2022 at a maximum of 12-day intervals during the period of operational Sentinel-1 SAR coverage. The uncertainty of these digitisations is defined to be the resolution of the Sentinel-1 Ground Range Detected (GRD) product (±10 m). Change in calving front position was calculated using a curvilinear box with a 1 km width using the MaQiT tool[74].

### Grounding line position
Measuring the grounding line position of Cadman Glacier in the period immediately prior to its acceleration and retreat is difficult because Sentinel-1 interferometric synthetic aperture radar (InSAR) is not coherent on the glacier, for example an automated InSAR processing and grounding line delineation campaign for 2018 was not able to resolve any grounding lines in this location[75]. Other remote sensing methods which rely on detecting tidal motion such as repeat track laser altimetry[76,77] and differential range offset tracking[78] do not adequately resolve Cadman Glacier. For example ICESat-2 tracks are widely spaced at this latitude and transverse to Cadman Glacier's flow direction, so do not cover the grounding zone. To locate the grounding line position in 2015, we measure the break in surface slope of the REMA 100 m Antarctic Peninsula DEM[64] between the ice shelf and grounded glacier[79]. We resample the REMA DEM to 10 m resolution using a cubic spline and delineate the grounding line at the most seaward local peak in surface slope gradient along the glacier's central flowline and extend this perpendicular to flow to the shear margins.

### Surface elevation change
We used altimetry data from the ICESat-2 laser altimeter, CryoSat-2 synthetic aperture radar interferometric (SARIn) altimeter, and stereo image DEMs to evaluate change in surface height and hence thinning of Cadman Ice Shelf and grounded inland ice. For ICESat-2, we use the ATL06 grounded ice elevation product to measure thinning of grounded ice from 2018 to 2022[80] using four repeats of track 361 from 21st April 2019, 19th Jan 2020, 17th April 2021, 17th August 2021. We differenced repeat measurements along the satellite track to calculate total surface elevation change and used a linear fit with all tracks to calculate thinning rates.

For CryoSat-2, we use Cryo-TEMPO radar altimetry data retrieved from the European Space Agency CryoSat-2 Science Server portal to evaluate the rate of surface lowering on Cadman Glacier's floating ice shelf[81,82]. There is insufficient swath-mode data to form a time-series, and so we use the Cryo-TEMPO Baseline B point of closest approach land ice (POCA_LI) product, selecting retrievals which are inside a box which covers the floating ice shelf (Figs. 1b, 4a). The altimetry datapoints are not consistently distributed within our sampling area between years, as the precessing orbit of CryoSat-2 means that exact repeat tracks only occur every 369-days, and POCA locations are rarely the same. To compute elevation changes from these non co-located measurements we correct for spatial variations in elevation of the glacier by referencing the CryoSat-2 data to the REMA DEM and measuring the difference between the CryoSat-2 elevation and the REMA elevation. We then calculated a surface elevation change rate for our whole sample area using a linear fit to these topographically corrected elevation differences.

We also performed a direct comparison between digital elevation models from different time periods to study changing surface elevation, using the Cook et al. (2012) 100 m Antarctic Peninsula DEM (Cook DEM)[83] and the REMA 100 m Antarctic Peninsula DEM[64]. The Cook DEM was produced as a corrected version of the ASTER Global Digital Elevation Map (A-GDEM) optimised for the AP. The A-GDEM is produced from ASTER stereo image tiles spanning 2000–2009. The REMA DEM is mosaiced from a large number of DEMs derived from several stereo-imaging satellites with a mean acquisition date of 9th May 2015 and a standard deviation of 432 days. We co-registered the Cook and REMA DEMs using the demcoreg.py package[84,85].

We used repeat 2 m resolution version 4.1 Reference Elevation Model of Antarctica (REMA) strips and the version 2 REMA composite Mosaic[50,64,86] to calculate ice thinning across the Cadman Glacier between November 2016 and February 2020. Specifically, the stereo imagery used to determine the DEM strips were acquired on 10/11/2016 and 25/02/2020. These were registered to the Mosaic over relatively stable surfaces, using the demcoreg.py package[84,85], with masks applied to remove surface water, clouds, blunders (using the binary mask provided with the REMA product); rapidly (> 75 m/yr) flowing ice[71], open ocean and floating ice[87]. By co-registering both DEMs to reference elevations over slow-flowing ice, and then differencing the two DEMs, we were thus able to estimate the anomalous dynamical thinning over the Cadman glacier relative to any background climatological signal.

## Ocean temperature and bathymetry

We used the GLORYS12V1 European Commission (EC) Copernicus Marine Service global ocean eddy-resolving reanalysis model (1/12° horizontal resolution, 50 vertical levels), to evaluate ocean temperature variability in the Bellingshausen Sea, between 1993 and 2020[88]. We define a 0.75° x 1.5° (approx. 70 km x 85 km) region of interest to evaluate changes in the properties of water masses which may access Cadman Glacier (Fig. 1a) and extract monthly potential temperature values at all model levels. To evaluate the temporal pattern of temperature anomalies, we calculate a monthly climatology for the region by averaging potential temperature for the period January 1993 to January 2010 at every grid point, and we then calculated a temperature anomaly by subtracting the climatology from potential temperature time-series. While ice thickness at the grounding line is highly uncertain on the AP due to the complex terrain and a paucity of bed elevation measurements, we aim to assess the temperature variability of ocean water in contact with Cadman Glacier's shelf and grounding zone.

We complement this ocean potential temperature reanalysis data by using direct ocean temperature and salinity measurements from CTD's from the Palmer Long-Term Ecological Research programme (LTER) campaign cruises on the west AP shelf. This dataset provides austral summer measurements of ocean physical properties through the water column at a regular grid of hydrographic stations on the west

AP shelf and in the shelf-adjacent ocean during cruises typically conducted in January and February cruises from 1993 to 2021. Stations sampled for this dataset are not fully consistent between years, therefore, given the highly localised nature of west AP shelf conditions we ensure fair comparison between years by restricting the dataset to 16 locations sampled for at least 20 of the 28 years. There are few direct oceanographic measurements in the LTER dataset close to Cadman Glacier, the nearest points that meet our sampling threshold are the Anvers-Hugo Trough (AHT) and Palmer Deep (PD), however, these locations are suitable for studying occurrences of warm water intrusion onto the shelf and onwards to Cadman Glacier. The flow of warm CDW onto the shelf is bathymetrically constrained and observations have shown evidence of preferential cross shelf transport in large bathymetric troughs[30,89]. Consequently, the AHT is an effective pathway for shelf-ocean exchange and likely for delivering heat to Cadman Glacier. The bathymetry of the west AP continental shelf is a product of past glaciations, with glacially carved throughs and over-deepened basins separated by shallow moraine sills[90–93]. These features control the flow of deep, warm ocean water toward the coast and are important for the modification of water masses through mixing[94]. For this reason, comparing temperatures between AHT and PD is useful to inform on the shoreward propagation of warm water masses, however, this is limited by a lack of measurements in the immediate vicinity of Beascochea Bay and Cadman Glacier.

The GLORYS reanalysis product assimilates the Coriolis Ocean Dataset for Reanalysis (CORA), which includes the World Ocean Database and at least part of the Palmer LTER dataset, so these data are not independent, and thus a formal validation is challenging. However, we compared the GLORYS reanalysis potential temperature data and measured potential temperature from the Palmer LTER profiles to confirm the reanalysis output reflects the variability of the in situ data (Fig. S3). We spatially resampled the monthly reanalysis data using linear interpolation onto the full set of sample locations for every LTER CTD in our dataset producing $1.3 \times 10^6$ sample points of measured and reanalysis potential temperature, comparing all points, we find a Pearson's correlation coefficient of 0.823 (Fig. S3). Our results show that the correlation between the two datasets is high and significant ($p < 0.05$) throughout the water column, with the magnitude varying with depth. We find that correlation is weakest in the 0–100 m and 100–200 m ranges, 0.672 and 0.593 respectively, with a positive bias in the reanalysis potential temperature in the 100–200 m interval, below the correlation improves significantly to 0.930 for depths below 200 m and 0.968 below 300 m (Fig. S3a–c); all correlation coefficients are significant ($p < 0.05$). Lower correlations higher in the water column may be expected because the reanalysis output is a monthly mean, while LTER CTDs are snapshots in time, so it is likely that higher frequency variability in surface waters is not well captured in the reanalysis product. Below 200 m, however, the reanalysis and measurements agree well; these depths are where the warmest waters are found and can cause enhanced glacier melt near the grounding line, therefore, we consider the reanalysis output of sufficient representativeness and appropriate for our investigation.

We extracted sea floor topography in the Hugo-Anvers Trough and Cadman Glacier fjord using direct bathymetric measurements from British Antarctic Survey vessels and data from the International Bathymetric Chart of the Southern Ocean (IBCSO v2)[95,96], supplemented by a previous study which mapped the geomorphology of Beascochea Bay from data collected in 2007[90].

## Data availability
Publicly available source data used in this study are available as follows: Copernicus Sentinel-1A/B and Sentinel-2A/B are available directly from the European Space Agency (https://scihub.copernicus.eu/, https://dataspace.copernicus.eu/); Landsat 5,7,8,9 imagery from USGS (https://glovis.usgs.gov/); ICESat-2 from Open Altimetry (https://openaltimetry.

org/data/icesat2/); Copernicus Marine Service GLORYS12V1 global ocean physics reanalysis data (https://doi.org/10.48670/moi-00021)[88]; CryoSat-2 Cryo-TEMPO POCA-LI (https://science-pds.cryosat.esa.int/#Cry0Sat2_data/TEMPO_POCA_LI); REMA Antarctic DEM V1 (https://doi.org/10.7910/DVN/SAIK8B); Cook et al. AP DEM: (https://doi.org/10.5194/essd-4-129-2012)[83]; International Bathymetric Chart of the Southern Ocean (https://ibcso.org/previous_releases/, https://doi.org/10.1002/grl.50413)[95,96]. Additional datasets and all datasets generated for this paper which are required to reproduce the results are available at (https://doi.org/10.5281/zenodo.10009820)[97]. This includes geometric definitions, ice velocity time-series, surface elevation change, calving front positions, oceanographic measurements, ocean reanalysis data, and subsets of the above publicly available data.

## Code availability

Ice velocity tracking was performed using GAMMA Remote Sensing software proprietary of GAMMA Remote Sensing, Switzerland. All data analysis is performed with MATLAB R2020b. Code used in this study including ice velocity post-processing code is available at (https://doi.org/10.5281/zenodo.10009820)[97]. Publicly available code used for this study can be found at: GEEDiT and MaQiT (https://liverpoolgee.wordpress.com/)[74], demcoreg.py (https://github.com/dshean/demcoreg)[84,85], mass discharge calculations (https://doi.org/10.5281/zenodo.7520043)[98], ImGRAFT (http://imgraft.glaciology.net/)[69].

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

## Acknowledgements

This work was led by the School of Earth and Environment at the University of Leeds. Data processing was undertaken on ARC3, part of the high-performance computing facilities at the University of Leeds, UK. The authors gratefully acknowledge the European Space Agency (ESA) for the provision of CryoSat-2 Cryo-TEMPO data, ESA and the European Commission (EC) for the acquisition and availability of Copernicus Sentinel-1 and Sentinel-2 data, and the EC for use of datasets produced through the Copernicus Marine Service. We acknowledge NASA and the U.S. Geological Survey for the acquisition and availability of Landsat and ICESat-2 data. We are grateful for the REMA DEMs provided by the Byrd Polar and Climate Research Center and the Polar Geospatial Center under NSF-OPP awards 1543501, 1810976, 1542736, 1559691, 1043681, 1541332, 0753663, 1548562, 1238993 and NASA award NNX10AN61G. Computer time provided through a Blue Waters Innovation Initiative. DEMs produced using data from Maxar. We also acknowledge J. Lea of the University of Liverpool for the public availability of the GEEDiT and MaQiT digitization tools. Alice Fremand (BAS) is thanked for provision of bathymetric data. The Palmer LTER program is supported by NSF-OPP Grant #2026045. Funding is provided to B.J.W. by the Panorama Natural Environment Research Council (NERC) Doctoral Training Partnership (DTP), under grant NE/S007458/1. A.E.H. was supported by the NERC DeCAdeS project (NE/T012757/1) and the ESA Polar+ Ice Shelves project (ESA-IPL-POE-EF-cb-LE-2019-834), A.E.H. and M.P.M. were supported by the ESA Polar+ SO-ICE project (ESA AO/1-10461/20/I-NB). Both ESA projects are part of the ESA Polar Science Cluster. M.P.M. was also supported by NERC award NE/W004933/1 (BIOPOLE), EU Horizon 2020 award 821001 (SO-CHIC) and EU Horizon Europe award 101060452 (OCEAN:ICE). R.C. was supported by a CASE studentship with the National Physical Laboratory, through the ENVISION Doctoral Training Partnership funded by the Natural Environment Research Council (NERC grant reference number NE/S007423/1). M.M. was supported by the NERC UK Centre for Polar Observation Modelling and the by the Lancaster University-UKCEH Centre of Excellence in Environmental Data Science. T.N. and J.W. acknowledge support from the European Space Agency through the ESA Polar+ Ice Shelves project and Antarctic Ice Sheet Climate Change Initiative (CCI +). C.M. was supported by NSF-OPP Grant #2026045. The authors thank Paul R. Holland of the British Antarctic Survey for helpful and insightful discussion on topics related to this manuscript.

## Author contributions

B.J.W. and A.E.H. designed this study. B.J.W. processed and analysed the ice velocity and calving front location data, except the TerraSAR-X velocity data which was processed by J.W. and T.N.; B.J.W. and C.M. prepared the oceanography data. B.J.W. and M.P.M. analysed the oceanography data. B.J.W. analysed the CryoSat-2 and ICESat-2 data. R.C., D.H., and M.M. analysed the repeat REMA DEM tiles. B.J.W. and A.E.H. wrote the manuscript. B.J.W, A.E.H., M.P.M., R.C., D.H., M.M., J.W., T.N., and C.M. contributed to scientific discussion, interpretation of the results and contributed to the manuscript.

## Competing interests

The authors declare no competing interests.
