## [Peer Review File · Nature Communications]

Ocean warming drives rapid dynamic activation of marine-terminating glacier on the west Antarctic PeninsulaREVIEWER COMMENTS

Reviewer #1 (Remarks to the Author):

Review of Wallis et al, Nature Comms

The authors marshal an impressive array of data sets to describe the recent evolution of Cadman Glacier, a major outlet of the northern section of the western side of the Antarctic Peninsula. The study focusses on the rapid thinning and acceleration that has occurred since about 2015, culminating in the loss of a floating ice tongue / ice shelf in 2021. The authors attribute the loss to a strong increase in basal melting beginning in 2017 or so as evidenced by detailed oceanographic data collected under the Palmer LTER program.

The study is well done and could be published after minor revisions and addressing a few notes and questions in the review. I will say that similar work has been reported numerous times for Greenland outlets by a host of researchers (Howat, Rignot, Joughin, Hamilton, Catania, Enderlin, Moon, Willis....) and this study details a very similar course of evolution for a fjord-bounded outlet glacier. However, this study is important in that the data sets are fairly comprehensive, span a significant period of time, and are temporally and spatially dense during the period of rapid evolution. Moreover, the breakup style of the relatively extensive ice tongue (relative to most Greenland outlets) appears to be unique - and more might be made of this fact: while hydrofracture processes and damage-related processes have dominated most literature on ice shelf break-up, the observation of a breakup with a significant component from basal melting (augmented by increased damage) may be new (not sure).

Several detailed comments are set as notes within the submitted .pdf file

There are valuable Landsat 8 images of the Cadman showing details of the loss of the ice shelf in Dec-Jan 2020 and 2021 that might illustrate the break-up mechanism, and could be added to your S1 figure. This style of break-up could be unique in the area (possible exception of the Wordie) in that melt ponding or hydrofracture does not appear to be involved. This strengthens your case that the ice shelf broke apart by a combination of increased damage due to acceleration and loss of contact with bedrock ridges near the evolving grounding line. Check 17 Jan 2020 and 26 Jan 2021 (and there are likely more) – to me it looks as though sub-ice shelf channels weakened the shelf by creating ~polygonal blocks that then concentrated strain and led to rapid calving along the channel lines. More Could Be Made Of This in your paper as an example of a style of ice shelf break-up that has not been widely discussed to date. Make note that your study does not show a presence of surface ponding (if that's the case; I think it is), a harbinger of hydrofracturing processes on a shelf.

Reviewer #2 (Remarks to the Author):

This paper investigates Cadman Glacier in the west Antarctic Peninsula (AP), focusing on its unique characteristics and recent changes. Unlike other glaciers in the region, Cadman Glacier velocity increased between 2018 and 2019 and experienced persistent thinning of its ice shelf. This thinning eventually led to retreat and rapid acceleration of the glacier. Warm ocean temperatures and bed geometry played significant roles in these changes. The study emphasizes the vulnerability of glaciers in the region to warm ocean waters and the need for further research to understand their future evolution.

The paper provides a comprehensive analysis of Cadman Glacier in the west Antarctic Peninsula, focusing on its unique characteristics and recent changes. The authors effectively present the background information and clearly articulate the research objectives. The methodology employed in the study, including the use of CryoSat-2 data and previous studies, appears to be partially appropriate for the research questions at hand.

The analysis of surface elevation, elevation change, and ice velocity provides valuable insights into the glacier's behavior, but grounding zone measurements are missing which could hinder the interpretation of the results. Additionally, I was surprised the authors did not provide both an eulerian and lagrangian melt rate analysis.

Overall, the findings of the study are significant and contribute to the understanding of glacier dynamics in the region. The observed persistent thinning of Cadman Glacier's ice shelf and its subsequent retreat and acceleration are compelling evidence of the glacier's vulnerability to warm ocean waters.

The paper effectively discusses the role of bed geometry and bathymetric features in influencing the glacier's response to environmental forcing. The inclusion of comparisons with neighboring glaciers adds depth to the analysis and supports the conclusions drawn. The implications of the study are well-discussed, particularly in terms of the impact on ocean circulation, productivity, and the ecosystem. The mention of potential regional-scale consequences further strengthens the significance of the research. The paper is well-written, with a clear and concise presentation of the research findings. The academic tone and language throughout the manuscript contribute to its overall quality.

However, one area that requires improvement is the discussion of limitations and future research directions. While the authors briefly mention the need for further studies and fieldwork measurements, more specific recommendations or suggestions for future investigations would enhance the paper's completeness specifically focused on the need of frequent grounding line measurements and melt rate measurements see (Shean et al 2019 Milillo et al 2022)

Additionally, it would be beneficial to provide a clearer connection between the findings of this study and their broader implications for climate change and sea-level rise. This would help contextualize the significance of the research within the larger scientific discourse.

Overall, this paper presents valuable research on Cadman Glacier and its response to environmental changes. With minor revisions and additions to the discussion section, this study has the potential to make a significant contribution to the field of glaciology and climate science.

Few more comments follows:

Unless a reference is missing, lines 73-75

seem a bit out of context for the introduction section and could be more appropriate in the conclusions

79-81 when referencing the 4 areas of the Cadman glacier a reference to a figure should guide the reader and enable the spatial identification of these areas on a map.

Fig1 missing reference to the DEM used for shading.

152-153 Which grounding line has been used to measure these proportions ?

Shean, D. E., Joughin, I. R., Dutrieux, P., Smith, B. E., & Berthier, E. (2019). Ice shelf basal melt rates from a high-resolution digital elevation model (DEM) record for Pine Island Glacier, Antarctica. *The Cryosphere*, 13(10), 2633-2656.

Milillo, P., Rignot, E., Rizzoli, P., Scheuchl, B., Mouginot, J., Bueso-Bello, J. L., ... & Dini, L. (2022). Rapid glacier retreat rates observed in West Antarctica. *Nature Geoscience*, 15(1), 48-53.

Reviewer #3 (Remarks to the Author):

This manuscript presents a detailed set of observations of Cadman Glacier, a marine-terminating glacier on the west coast of the Antarctic Peninsular, spanning a period of rapid change in the glacier's ice shelf. The observations, which include newly derived data on ice thickness, velocity and terminus position, clearly document the thinning, ungrounding, acceleration, retreat and ultimate collapse of the ~5 km ice shelf over the period ~2010-2022. Through comparison with existing hydrographic and bathymetric data, the authors present a convincing case that this reflects the influence of the ocean, with gradual weakening of the shelf due to long term thinning followed by a final collapse triggered by the presence of particularly warm ocean waters on the shelf. Altogether, this makes a nice case study of a moderately sized marine terminating glacier experiencing ice shelf thinning and collapse in response to warming ocean waters. The detailed observations document this process well, and I believe the methodology, analysis and interpretation are sound.

While I am confident that the work is worthy of publication in a relevant journal, I am less confident that it represents the kind of major, high-impact advance that would typically be associated with *Nature Communications*. The finding that the retreat of glaciers on the western Antarctic Peninsular is predominantly ocean-driven was presented on a much larger scale by Cook et al (2016) – the current manuscript presents a detailed case study of one such example of this, but does not to me really advance our broader understanding of this topic beyond this particular glacier (which is not in itself of

particular importance as a source of sea level rise etc). The sequence of processes observed - ice shelf thinning, acceleration and break up, grounding line retreat, dynamic thinning upstream – are quite familiar from the loss of other ice shelves and tongues in Antarctica and Greenland (e.g. Motyka et al (2011), to give just one example). The attribution to oceanic forcing supports the findings of Cook et al (2016), but the study lacks the kind of novel observations (such as in situ hydrographic measurements close to the glacier, or high resolution / spatially resolved measurements of shelf thinning) that might really advance our understanding of the underlying processes.

Putting this aspect aside, I do not have any major concerns with the methods, analysis or interpretation as currently presented. I found the manuscript to be largely well written, if perhaps overly descriptive at times – it might be possible to make better use of the figures and reduce the amount of descriptive text a little. I have a few more minor comments, outlined below.

L28. The 'cryosphere' is generally used to refer to ice and snow on a global scale – better to refer to the glaciers of this region or similar

L37. Ambiguous phrasing – suggest something like 'increased by 400 % between the periods 1992-1997 and 2007-2012'

L161. A figure reference would be useful

L175-185. It would be useful to add a reminder of the years of the ASAD grounding line and DEMs

L244-5. It's not easy to discern the grey dots in the figure, perhaps a different colour would work better

L264-275. It wasn't clear to me why the addition of surface lowering merited such attention – is this making a methodological point (that other researchers need to make sure they factor this in), or that surface lowering has an important impact on the dynamics of the glacier?

L303-306. Yes these glaciers have shallower sills, but its notable in Figures 6c-d that there is still substantial warming at depths shallower than 200-230 m, so they should still have experienced significantly warmer temperatures at this time.

L311-312. What year is the ocean temperature anomaly shown?

L326-328. It would be helpful to include a figure reference

L328-329. What is the postulated driver of this long term thinning? Ocean warming?

L371-373. Again it would be helpful to include a figure reference

References

Cook, A., Holland, P., Meredith, M., Murray, T., Luckman, A., & Vaughan, D. (2016). Ocean forcing of glacier retreat in the western Antarctic Peninsula. *Science*, 353(6296), 283-286.

Motyka, R. J., Truffer, M., Fahnestock, M., Mortensen, J., Rysgaard, S., & Howat, I. (2011). Submarine melting of the 1985 Jakobshavn Isbrae floating tongue and the triggering of the current retreat. *Journal of Geophysical Research-Earth Surface*, 116, Article F01007. <https://doi.org/10.1029/2009jf001632>

Response to reviewers: Ocean warming drives rapid dynamic activation of a marine-terminating glacier on the west Antarctic Peninsula

Ref: NCOMMS-23-19496

Reviewer 1:

Ref.	Original Line	Comment	Revised Line	Response
1.1		The authors marshal an impressive array of data sets to describe the recent evolution of Cadman Glacier, a major outlet of the northern section of the western side of the Antarctic Peninsula. The study focusses on the rapid thinning and acceleration that has occurred since about 2015, culminating in the loss of a floating ice tongue / ice shelf in 2021. The authors attribute the loss to a strong increase in basal melting beginning in 2017 or so as evidenced by detailed oceanographic data collected under the Palmer LTER program. The study is well done and could be published after minor revisions and addressing a few notes and questions in the review. I will say that similar work has been reported numerous times for Greenland outlets by a host of researchers (Howat, Rignot, Joughin, Hamilton, Catania, Enderlin, Moon, Willis...) and this study details a very similar course of evolution for a fjord-bounded outlet glacier. However, this study is important in that the data sets are fairly comprehensive, span a significant period of time, and are temporally and spatially dense during the period of rapid evolution. Moreover, the breakup style of the relatively extensive ice tongue (relative to most Greenland outlets) appears to be unique - and more might be made of this fact: while hydrofracture processes and damage-related processes have dominated most literature on ice shelf break-up, the observation of a breakup with a significant component from basal melting (augmented by increased damage) may be new (not sure). Several detailed comments are set as notes within the submitted .pdf file		We thank the reviewer for their comments regarding the quality and importance of the study, and for their insightful, informative and productive comments as a whole. In particular we thank the reviewer for taking the time to independently study satellite images of Cadman Glacier and provide helpful suggestions based on these. We have replied to each comment in turn and made changes to the manuscript to address them. A major suggestion by the reviewer was to include more optical satellite images of Cadman Glacier in figure S1. We have done this and expanded the discussion of the failure mechanism for the Cadman Ice Shelf. We agree with the reviewer that the importance of this wasn't highlighted clearly enough in the previous version. We believe that changes and clarifications presented in the revised manuscript have addressed the reviewer's concerns and have enriched the manuscript. We thank the reviewer for their time and effort.
1.2		There are valuable Landsat 8 images of the Cadman showing details of the loss of the ice shelf in Dec-Jan 2020 and 2021 that might illustrate the break-up mechanism, and could be	120-123 128-130 361-365	We thank the reviewer for this helpful suggestion, which we agree would add a further interesting perspective to the manuscript.

		added to your S1 figure. This style of break-up could be unique in the area (possible exception of the Wordie) in that melt ponding or hydrofracture does not appear to be involved. This strengthens your case that the ice shelf broke apart by a combination of increased damage due to acceleration and loss of contact with bedrock ridges near the evolving grounding line. Check 17 Jan 2020 and 26 Jan 2021 (and there are likely more) – to me it looks as though sub-ice shelf channels weakened the shelf by creating ~polygonal blocks that then concentrated strain and led to rapid calving along the channel lines. More Could Be Made Of This in your paper as an example of a style of ice shelf break-up that has not been widely discussed to date. Make note that your study does not show a presence of surface ponding (if that's the case; I think it is), a harbinger of hydrofracturing processes on a shelf.		Done – we have expanded Figure S1 to include the 2020/01/17 Landsat-8 image that the reviewer suggested and an additional image of the Cadman fjord after the ice shelf collapse. We have changed the description of damage on the shelf to describe this: 120: 'Although advanced, during this period the seaward 3 km of glacier tongue shows a substantial increase in damage and crevassing on the ice shelf surface and a widening of the shear margins (Fig. S1), suggesting that the structural integrity of the ice was decreasing. Notably in the 17th January 2020 image (Fig. S1d) there is a distinctive along-flow damage feature on the central ice shelf, which may indicate channelised sub-shelf melting.' And we have expanded the discussion of the lack of surface melt ponds throughout the manuscript: 128: 'Notably, although summer melt is widespread in the region, we observe no surface meltwater ponds visible on the ice surface in any optical satellite image for our period of study (1984 -2021), which we attribute to the north-west AP's thick firn layer and high firn air content.' 361: 'The Cadman Ice Shelf fully collapsed in March 2021 due to this thinning, possibly intensified through channelised melting, and the loss of pinning on the 2015 ridge. The absence of surface melt ponding on the Cadman Ice Shelf prior to its collapse demonstrates that surface ponding is not a necessary precondition for ice shelf collapse, which can occur through ocean-driven thinning and unpinning alone.'
1.3	15	please re-write this sentence! These types of statements are meaningless, and a bit disrespectful. Don't we have a very good idea of the processes involved? Don't you cite that work as the basis for your work? Something more positive perhaps - instead of the 'however' phrase, maybe 'and detailed studies of areas with significant ongoing change inform and improve models for better forecasting.'	14	Done – changed to: 14: 'Ice dynamic change is the primary cause of mass loss from the Antarctic Ice Sheet, thus it is important to understand the processes driving ice-ocean interactions and the timescale on which major change can occur.' Apologies if there was a perception of disrespect which was certainly not our intention.
1.4	16	remove 'on' -- 'fronting' or 'at the terminus of'	16	Done – changed to 'fronting'
1.5	47	remove 'from the deeper layers of' a bit confusing... the continental shelf intrusion of warm water comes from the upper- layer of the ACC, but its deeper than the cooler Bransfield Strait water (I believe so).	47	Done – changed to: 47: '...transported by the Antarctic Circumpolar Current and intrudes onto the continental shelf from the open ocean'
1.6	47	insert: Antarctic continental shelf	47	Done – changed to 'continental shelf', we believe Antarctic is clear from context.

1.7	54	change to -- The western flank of the AP has very high accumulation, due to orographically driven precipitation and the strong westerly winds	54	Done – changed to: 54: ‘The west AP has high accumulation, due to orographically driven precipitation and the strong westerly winds’
1.8	55	change 'air pocketed' to 'porous'.	56	Done – much better wording, thank you
1.9	56	please add reference: Montgomery, L., Miège, C., Miller, J., Scambos, T.A., Wallin, B., Miller, O., Solomon, D.K., Forster, R. and Koenig, L., 2020. Hydrologic properties of a highly permeable firm aquifer in the Wilkins Ice Shelf, Antarctica. Geophysical Research Letters, 47(22), p.e2020GL089552. This is the discovery paper for firm aquifers.	56	Done – reference added
1.10	58	I don't think heat loss to the atmosphere directly affects the mCDW layer - mixing with the polar water layer may be said to be induced by heat loss to the atmosphere, but this should be rephrased.	57	We agree this wording should be refined. Done – rephrased to 57: ‘...becoming modified CDW (mCDW) through mixing and vertical heat loss to the overlying Antarctic Surface Water layer’ this matches the description used by Moffat & Meredith, 2018.
1.11	58	add: ',both shallow and deep, '	59	Refers to: This contrasts with the much colder waters of the Bransfield Strait and Weddell Sea Comment – We feel this wording would make this sentence less clear, so have left it unchanged.
1.12	61	... ,a large area of sea ice persists during austral summers. becoming....	61	Done – changed to: 61: ‘...a large area of sea-ice regularly persists during austral summer’
1.13	62	This should not be characterized as a monotonic increase spanning 1979 through 2015; there was a strong reduction in the NW Weddell from the late 1980s through the early 2000s, but an erratic increasing trend on either side of that time-span	62	Done – we have rephrased this as: 62: ‘Between 1978 and 2015 sea-ice extent in the Weddell Sea grew with a small positive trend superimposed upon large interannual variability’ We have also added a reference to Turner et al., 2020, which describes the Weddell Sea sea-ice conditions in detail.
1.14	64	this was surpassed in 2022 and again in 2023.	63	For Weddell Sea specific sea ice extent this may not be true, however a negative anomaly did contribute to the total Antarctic record low in 2022. Done – rephrased and added reference to (Turner et al., 2022) 63: ‘...from 2016 onwards this trend reversed with satellite data showing a reduction in SIE, reaching a record low in 2019 and contributing to record low total Antarctic SIE in 2022’

1.15	66	please add comma -- ...precipitation, snow, and glacier melt...	67	Done
1.16	80	this does not read like a 'definition' of the lower glacier	79	Done – changed to ‘we define’ as this is a naming we have chosen. We chose to call this part lower as it is below the Cadman icefall which separates the upper slow flowing basin from this lower section.
1.17	94	please spell this out a bit more -- accelerated by 94%, a speed-up of 1.5 km/yr, reaching (something like) 2.9 km/yr by xx date.	93	Comment – This short paragraph is just to introduce the reader to the acceleration event being discussed, so that the following paragraphs are not out of context. We discuss the ice speed changes in detail in the results section so we would prefer not to duplicate such detail in the introduction.
1.18	129	There are valuable images of the loss of the ice shelf in Dec-Jan 2020 and 2021 that appear to illustrate the break-up mechanism	120-129	Done – see response above to point 1.2
1.19	Fig 3	add a note or symbol here that this reference is the grounding line in the figure -- saves time in understanding what is what here.	Fig 3	Done – changed to: ‘given with respect to the ASAID grounding line location (illustrated in Fig 1b)’
1.20	217	I think you should consider the possibility that this feature, the area near the peak of elevation loss, is related to a sub-glacial lake behind the ridge that was induced to drain by changes in the glacier and therefore changes in the hydrostatic gradient at the ice base. Papers by Gary Clarke discuss these sorts of possibilities.	217	Comment – This is a really interesting suggestion and something we will consider in future work, but we feel it would be too speculative to include here.
1.21	239	I would add 'and loss of basal stress over the ridge region'	244	Done – changed to: 244: ‘The coincident timing of these thinning observations with the ice speed change measurements demonstrates that this is a dynamically induced thinning caused by the 2018/19 ice acceleration, which was in turn caused by the loss of basal traction over the 2015 bed ridge.’
1.22	282	Split the sentence here	287	Done
1.23	331	its not really 'damage' in the sense that that is usually discussed - it is a weakening and incising by basal melting (I think.... more images are available of the events)	338	Comment – We use ‘damage’ as an inclusive term to describe fracture and weakening.
1.24	409	change 'chance' -- more like: '...represents a uniquely detailed assemblage of data, allowing us to characterize the transition of an Antarctic...'	432	Done – changed to: 432: ‘This study represents a uniquely detailed assembly of data allowing us to observe and characterise the forced transition of an Antarctic glacier’
1.25	431	just saying: have a look at ITS_LIVE data set and the other MEaSURES tools		Comment - The ITS_LIVE online data tool timeseries begins in 2014 for points sampled at Cadman glacier and includes only a small number of additional optically tracking results

				during this period, so it is broadly comparable to our own dataset. However, it was a useful exercise to confirm that ITS_LIVE measures the same acceleration of Cadman as we observe in this study.
--	--	--	--	--

Reviewer 2

Ref.	Original Line	Comment	Revised Line	Response
2.1		This paper investigates Cadman Glacier in the west Antarctic Peninsula (AP), focusing on its unique characteristics and recent changes. Unlike other glaciers in the region, Cadman Glacier velocity increased between 2018 and 2019 and experienced persistent thinning of its ice shelf. This thinning eventually led to retreat and rapid acceleration of the glacier. Warm ocean temperatures and bed geometry played significant roles in these changes. The study emphasizes the vulnerability of glaciers in the region to warm ocean waters and the need for further research to understand their future evolution.		We thank the reviewer for their comments and constructive feedback. In particular we were pleased to hear the reviewer found the study had the potential to make a significant contribution to the field of glaciology. We have replied to the reviewers comments in turn and made changes to the manuscript to address them. Specifically, we have added an updated grounding line measurement, including details of the method used. We justify why we do not feel it is appropriate to include a Lagrangian basal melt rate analysis in the manuscript, through comparison to other studies and considerations relating to the other datasets required for this calculation. Finally, we have significantly expanded the discussion section of the manuscript to provide concrete recommendations for future work and to place the results in the greater context of the Antarctic Peninsula region's contribution to rising sea-levels. Overall, we believe that addressing the reviewer's comments has substantially enhanced the manuscript. We thank the reviewer for their time and consideration.
2.2		The paper provides a comprehensive analysis of Cadman Glacier in the west Antarctic Peninsula, focusing on its unique characteristics and recent changes. The authors effectively present the background information and clearly articulate the research objectives. The methodology employed in the study, including the use of CryoSat-2 data and previous studies, appears to be partially appropriate for the research questions at hand. The analysis of surface elevation, elevation change, and ice velocity provides valuable insights into the glacier's behavior, but grounding zone measurements are missing which could hinder the interpretation of the results.	211-212 345-351 491-501 Fig. 1b Fig. 3b	Grounding line: Directly measuring the grounding line position of Cadman Glacier is difficult because there is no InSAR coherence on the glacier in available datasets. For example, a study which processed all available Sentinel-1 data in Antarctica for 2018 and automatically delineated grounding lines did not produce any measurements for Cadman Glacier (Mohajerani et al., 2021). We investigated measuring grounding line position using repeat track ICESat-2 laser altimetry (Fricker and Padman, 2006; Brunt et al., 2010), but found insufficient data coverage for the Cadman grounding zone, due to the large spacing between tracks at lower latitudes and an unfavorable across-glacier track orientation. In the original manuscript, we discuss that the grounding line of Cadman glacier likely retreated from the ASAD dataset (1999-2003) position to a position around 2km inland based on changes in ice flow speed and DEMs: Original MS L340-344: 'We interpret these changes to be indicators of a loss of basal

			traction and therefore the retreat of the grounding line, from the AS Aid grounding line location to a position 2 km further inland which is co-located near the point (ii) 2015 surface bump and associated speed minimum (Fig. 3).' While a precise InSAR grounding line measurement is not possible, we have formalised our analysis of the pre-acceleration grounding line position by measuring the break in surface slope of the 2015 REMA DEM (Howat et al., 2019). We define this as the most seaward local peak in surface slope gradient (Hogg et al., 2018; Friedl et al., 2020) and find the grounding line position to be 1.25 km inland of the AS Aid (1999 – 2003) position. Done – we have amended the text in several places to include this measurement. In the methods section: 491: 'Measuring the grounding line position of Cadman Glacier in the period immediately prior to its acceleration and retreat is difficult because Sentinel-1 interferometric synthetic aperture radar (InSAR) is not coherent on the glacier, for example an automated InSAR processing and grounding line delineation campaign for 2018 was not able to resolve any grounding lines in this location. Other remote sensing methods which rely on detecting tidal motion such as repeat track laser altimetry and differential range offset tracking do not adequately resolve Cadman Glacier. For example ICESat-2 tracks are widely spaced at this latitude and transverse to Cadman Glacier's flow direction, so do not cover the grounding zone. To locate the grounding line position in 2015, we measure the break in surface slope of the REMA 100 m Antarctic Peninsula DEM between the ice shelf and grounded glacier. We resample the REMA DEM to 10 m resolution using a cubic spline and delineate the grounding line at the most seaward local peak in surface slope gradient along the glacier's central flowline and extend this perpendicular to flow to the shear margins.' In the surface elevation change results section: 211: '...the break in surface slope in REMA DEM as an approximation for location of Cadman Glacier's grounding line in 2015, we find that in 2015 the grounding line is 1.25 km inland of the AS Aid 1999-2003 position (Fig 3b).' In the discussion: 345: 'Between 1991 and 2010 there was a change from a local ice speed minimum feature to an area of spatially constant ice speed immediately behind the AS Aid grounding line location, which was matched by the loss of an ice surface bump between the Cook AP (2000-2009) and the REMA DEMs (2015), together with the inland migration of the break in surface slope. We interpret these changes to be indicators of a loss of basal traction and the retreat of the grounding line, from the AS Aid grounding line location to a position 1.25 km further inland
--	--	--	---

			located near the point (ii) 2015 surface bump and associated speed minimum (Fig. 3). Such a loss of basal traction could also explain the increase in ice speed of 0.25 km/yr (15 %) between 1991 and 2010.' We have also updated Figure 1b and Figure 3b to show the 2015 grounding line position.
2.3	Additionally, I was surprised the authors did not provide both an eulerian and lagrangian melt rate analysis.		Basal melt rates: The reviewer raises a question about why both Eulerian and Lagrangian basal melt rate calculations were not included. Note that we did not calculate a Eulerian or Lagrangian basal melt rate value for the Cadman Ice Shelf in the original manuscript, but instead calculated a total ice shelf thinning rate for the shelf area defined in Fig 1b and Fig 4a, which has an area of 9.3 km². This was a deliberate choice we made because we did not feel that the Cadman Ice Shelf was adequately resolved in the additional datasets required to calculate a basal melt rate. Specifically, for surface mass balance (SMB), the highest spatial resolution regional climate model available to us is the 5.5 km RACMO 2.3p2 Antarctic Peninsula model (van Wessem et al., 2016) and for firn air content (FAC), existing products, such as those produced by the IMAU firn densification model have a resolution of 27 km. The width of the Cadman ice shelf is approximately 3 km, therefore it would not be well resolved in either of these datasets. The shelf is surrounded by steep topography up to 2 km high, meaning the values for SMB and FAC in these models for the single grid cell containing the Cadman ice shelf will likely not be representative of conditions on the shelf. Furthermore, for any Lagrangian calculation individual altimetry results must be advected with ice flow and an elevation change rate calculated via a plane fit to an appropriate grid postings resolution. Grid resolutions for altimetry based basal melt data in the literature vary from 500 m (Gourmelen et al., 2017) to 30 km (Adusumilli et al., 2018), with the finer grid resolution calculations using high resolution swath mode CryoSat-2 altimetry which was not available on Cadman Ice Shelf. Additionally, the ice shelves to which Lagrangian calculations have been applied are many orders of magnitude larger than the Cadman ice shelf. Overall, we believe these methodological limitations prevent the accurate calculation of a useful and robust basal melt rate value, which justifies our choice to use a Eulerian average ice shelf thinning rate which we can report with confidence.
2.4	Overall, the findings of the study are significant and contribute to the understanding of glacier dynamics in the region. The observed persistent thinning of Cadman Glacier's ice shelf		No change required, thank you for the positive assessment.

		and its subsequent retreat and acceleration are compelling evidence of the glacier's vulnerability to warm ocean waters.		
2.5		The paper effectively discusses the role of bed geometry and bathymetric features in influencing the glacier's response to environmental forcing. The inclusion of comparisons with neighboring glaciers adds depth to the analysis and supports the conclusions drawn. The implications of the study are well-discussed, particularly in terms of the impact on ocean circulation, productivity, and the ecosystem. The mention of potential regional-scale consequences further strengthens the significance of the research. The paper is well-written, with a clear and concise presentation of the research findings. The academic tone and language throughout the manuscript contribute to its overall quality.		No change required
2.6		However, one area that requires improvement is the discussion of limitations and future research directions. While the authors briefly mention the need for further studies and fieldwork measurements, more specific recommendations or suggestions for future investigations would enhance the paper's completeness specifically focused on the need of frequent grounding line measurements and melt rate measurements see (Shean et al 2019 Milillo et al 2022)	433-451	Done – Added the following to the discussion 433: 'This study represents a uniquely detailed assembly of data allowing us to observe and characterise the forced transition of an Antarctic glacier from one with a small floating shelf to a tidewater glacier. It highlights the impact of continuous, high-resolution multi-decadal climate observations and remote sensing campaigns which have contributed to our understanding of the complex interaction of Earth-system processes and how Antarctica will respond to a warmer climate. This work also shows the importance of continuing and improving these observations. Dense time-series of ice velocity measurements, such as those from Sentinel-1, could be further enhanced by dedicated campaigns of in-situ validation to quantify the impact of surface processes on velocity measurement error, allowing greater sensitivity to detect short-term ice dynamic changes. Substantial further insights would also be gained from frequent grounding line position measurements, particularly in regions such as the AP where low SAR coherence currently makes these measurements difficult. Improving the temporal and spatial resolution of altimetry measurements will improve the monitoring of ice shelf thinning across the Antarctic continent. Expanded airborne ground penetrating radar campaigns to measure ice thickness and bed topography near the grounding zone would reduce uncertainty in ice discharge and better characterise which glaciers are vulnerable to dynamic tipping points like the one observed in this study. Furthermore, Cadman Glacier and the Beascochea Bay system including Funk and Lever Glaciers are strong candidates for further studies and fieldwork measurements. New and sustained oceanographic measurements of the

				bay and individual glacier fjords behind the bathymetric sills would be valuable to understand heat transport across the west AP shelf and to compare circulation between Greenlandic fjords and the AP helping to better understand the future evolution of the Antarctic Peninsula's glaciers.'
2.7		Additionally, it would be beneficial to provide a clearer connection between the findings of this study and their broader implications for climate change and sea-level rise. This would help contextualize the significance of the research within the larger scientific discourse.	361-365 415-425	Done – we have extended the discussion of the broader implications of this study and we have expanded a number of points in the discussion. On how this study presents an example of ice shelf collapse without surface ponding (also see comments to reviewer 1, comment 1.3): 361: 'The Cadman Ice Shelf fully collapsed in March 2021 due to this thinning, possibly intensified through channelised melting, and the loss of pinning on the 2015 ridge. The absence of surface melt ponding on the Cadman Ice Shelf prior to its collapse demonstrates that surface ponding is not a necessary precondition for ice shelf collapse, which can occur through ocean-driven thinning and unpinning alone.' Describing the broader implications for sea-level rise and climate: 414: 'Our results demonstrate that such tipping points can increase ice discharge extremely rapidly, in Cadman Glacier's case by 28.1% in 13 months. Other glaciers on the Antarctic Peninsula may be vulnerable to similar sudden acceleration and retreat, where bed conditions and bathymetry allow, similar to heterogeneous patterns of tidewater glacier retreat controlled by bed topography observed in Greenland. For example, the largest glacier on the west coast, Fleming Glacier, has a floating tongue and is grounded on a retrograde bed slope, similar to Cadman Glacier. Despite the relatively small size of the Antarctic Peninsula Ice Sheet compared to the West and East Antarctic Ice Sheets, the stability of the AP's glaciers is important to global sea-level. The region has contributed 1.4 ± 0.37 mm to global mean sea-level from 1992 to 2017, 19% of the Antarctic total. Additionally, increased freshwater input from ice discharge and newly exposed seabed due to glacier retreat would have regional-scale consequences for ocean circulation, productivity, carbon sequestration, and the ecosystem.'
2.8		Overall, this paper presents valuable research on Cadman Glacier and its response to environmental changes. With minor revisions and additions to the discussion section, this study has the potential to make a significant contribution to the field of glaciology and climate science.		Done – Thank you for the comment. No change required.
2.9	73-75	Unless a reference is missing, lines 73-75 seem a bit out of context for the introduction section and could be more appropriate in the conclusions	431	Done – Removed this from here and made point in discussion where it has better context, see response to 2.6 above.

2.10	79-81, Fig. 1	when referencing the 4 areas of the Cadman glacier a reference to a figure should guide the reader and enable the spatial identification of these areas on a map.	77-79	Comment – This description is intended to familiarize the reader with the glacier and the definitions here are only used qualitatively for description. Quantitative geometric definitions and those used for calculations are defined on a map in Figure 1b. We feel adding more labels to Figure 1b would reduce clarity, so have left it unchanged.
2.11	Fig. 1	Fig1 missing reference to the DEM used for shading.	Fig. 1	No action required – Figure 1b,c,d all use the Landsat-8 image referenced in the caption as a background, no DEM was used.
2.12	152-153	Which grounding line has been used to measure these proportions ?	152	This relates to the sample areas defined in Figure 1b and Figure 3b. Done - Added detail to the introductory sentence: 152: 'Our 30-year timeseries of ice velocity (Fig. 2, Fig. 3a, sample areas defined Fig 1b, 3b)'

Reviewer 3

Ref.	Original Line	Comment	Revised Line	Response
3.1		This manuscript presents a detailed set of observations of Cadman Glacier, a marine-terminating glacier on the west coast of the Antarctic Peninsular, spanning a period of rapid change in the glacier's ice shelf. The observations, which include newly derived data on ice thickness, velocity and terminus position, clearly document the thinning, ungrounding, acceleration, retreat and ultimate collapse of the ~5 km ice shelf over the period ~2010-2022. Through comparison with existing hydrographic and bathymetric data, the authors present a convincing case that this reflects the influence of the ocean, with gradual weakening of the shelf due to long term thinning followed by a final collapse triggered by the presence of particularly warm ocean waters on the shelf. Altogether, this makes a nice case study of a moderately sized marine terminating glacier experiencing ice shelf thinning and collapse in response to warming ocean waters. The detailed observations document this process well, and I believe the methodology, analysis and interpretation are sound.		Comment - We thank the reviewer for their consideration of the manuscript and helpful comments. We are please that the reviewer found the work to be robust and well written. We thank the reviewer for their time. Done - We have addressed the reviewers comment point by point below. Specifically, the reviewer's comment about sill depth prompted us to expand the discussion of how fjord sills influence circulation and basal melt rates, which we feel notably improved this section of the manuscript.
3.2		While I am confident that the work is worthy of publication in a relevant journal, I am less confident that it represents the kind of major, high-impact advance that would typically be associated with Nature		Comment: We feel the results presented in this study will be of great interest to scientists from a broad range of disciplines, which is why we chose to submit to Nature Communications. Our work combines high resolution remote sensing data with direct oceanographic measurements

		Communications. The finding that the retreat of glaciers on the western Antarctic Peninsular is predominantly ocean-driven was presented on a much larger scale by Cook et al (2016) – the current manuscript presents a detailed case study of one such example of this, but does not to me really advance our broader understanding of this topic beyond this particular glacier (which is not in itself of particular importance as a source of sea level rise etc). The sequence of processes observed - ice shelf thinning, acceleration and break up, grounding line retreat, dynamic thinning upstream – are quite familiar from the loss of other ice shelves and tongues in Antarctica and Greenland (e.g. Motyka et al (2011), to give just one example). The attribution to oceanic forcing supports the findings of Cook et al (2016), but the study lacks the kind of novel observations (such as in situ hydrographic measurements close to the glacier, or high resolution / spatially resolved measurements of shelf thinning) that might really advance our understanding of the underlying processes.		and reanalysis to significantly develop the process-based link between ice-ocean interactions in Antarctica, which is at the forefront of modern climate science and is essential for accurately projecting future sea-level rise. In response to the comments of Reviewer 1, we have provided more detail about the novelty of the break-up of Cadman Ice Shelf without melt ponding and hydrofracture, a point which was not emphasized sufficiently in the original manuscript. See response to point 1.2. Furthermore, through the responses to this point and those raised by other reviewers, we have substantially expanded the discussion to stress how the results of this study relate to broader research questions regarding ice-ocean interactions in Antarctica. See response to point 2.7. We also expanded our discussion of recommendations for future work and highlight the remaining challenges which must be overcome to better understand the glacier dynamics of this region. See response to point 2.6. The reviewer is correct to cite the link between ocean warming and glacier retreat on the west AP found by Cook et al. 2016. We further develop this link to show that, as well as glacier retreat, ocean warming also causes increased ice discharge and mass loss, which directly contributes to sea-level rise. We note that since Cook et al.'s 2016, there have been few further papers exploring this link through observations. Compared to Greenland, the AP is a sparsely observed, but still important region making up 19% of Antarctica's sea-level contribution. For these reasons we believe our work will be of significant interest to the cryosphere, climate, and ocean science communities. We hope the reviewer agrees that the changes made in response to all the reviewers' comment have improved the manuscript, further highlighted the important scientific results, and provide improved interest for the broad audience that Nature Communications commands. Of course, we respect the editor's final decision on the suitability of this manuscript for Nature Communications.
3.3		Putting this aspect aside, I do not have any major concerns with the methods, analysis or interpretation as currently presented. I found the manuscript to be largely well written, if perhaps overly descriptive at times – it might be possible to make better use of the figures and reduce the amount of descriptive text a little. A have a few more minor comments, outlined below.		No change required.
3.4	28	The 'cryosphere' is generally used to refer to ice and snow on a global scale	27	Done – changed to:

		– better to refer to the glaciers of this region or similar		27: ‘major changes in the region’s glaciers and ice shelves throughout the 20th and 21st centuries’
3.5	37	Ambiguous phrasing – suggest something like ‘increased by 400 % between the periods 1992-1997 and 2007-2012’	37	Done - changed to: 37: ‘400 % between the periods 1992 to 1997 (7 ± 13 Gt/yr) and 2007 to 2012 (35 ± 17 Gt/yr)’
3.6	161	A figure reference would be useful	166	Done – added reference to Fig. 3a
3.7	175-185	It would be useful to add a reminder of the years of the ASAIID grounding line and DEMs	Fig. 3b	Done – the DEM years are already referenced in the legend for Fig 3b, but we have added the ASAIID years to the figure caption.
3.8	244-245, Fig. 4	It’s not easy to discern the grey dots in the figure, perhaps a different colour would work better	Fig. 4	Done – we added a black border to these dots to make them clearer and removed the lower glacier sampling area (red dashed) which was not relevant to this plot
3.9	264-275	It wasn’t clear to me why the addition of surface lowering merited such attention – is this making a methodological point (that other researchers need to make sure they factor this in), or that surface lowering has an important impact on the dynamics of the glacier?	378-382	The surface lowering term is significant because it reduces the ice discharge over time, even if surface velocities remain high. It is one of the ways that the glacier system can return to mass budget equilibrium. By accounting for this term, we can estimate the time taken for the glacier to return to equilibrium. This point is expanded upon in the discussion, lines 374-378. Done – we have amended the discussion section to make this link clearer: 378: ‘Although ice discharge has decreased from its post-acceleration peak due to dynamic thinning, it is unclear when the glacier reaches a new equilibrium, or even if it can reach a new equilibrium should environmental forcing change faster than the glacier’s response time. However, even with ice speed unchanged from 2022 values, at the current rate of dynamic thinning (-20.1 ± 2.6 m/yr across our flux gate) ice discharge will return to its 2016-2019 mean by October 2024.’
3.10	303-306	Yes these glaciers have shallower sills, but its notable in Figures 6c-d that there is still substantial warming at depths shallower than 200-230 m, so they should still have experienced significantly warmer temperatures at this time.	397-404	Comment - We thank the reviewer for raising this point and agree that the original discussion around the influence of the fjord sills could be expanded to be more comprehensive and nuanced. The oceanographic measurements we have used, while very interesting, are not direct measurements of the water properties in the individual glaciers’ fjords, as they are taken from points on the west AP shelf and we are careful to note this clearly in the paper. Recent observations in north Greenland and model studies show that fjord sills seaward of the grounding line of ice tongue glaciers exert a hydraulic control on buoyant meltwater plume driven fjord circulation (Jakobsson et al. , 2020; Schaffer et al. , 2020; Nilsson et al. , 2022; Bao and Moffat, 2023). Conceptually, this can be understood as two regimes, one with low or insignificant sills which is melt-controlled and one hydraulically-controlled, where sills

				constrain exchange circulation. Importantly, modelling shows that sills do not need to be high enough to fully block warm deep water to substantially reduce melt rates at the grounding line. In this framework Cadman Glacier would be a melt-controlled system while neighboring Funk and Lever would be hydraulically controlled. Without direct oceanographic measurements it is not possible to definitively know the circulation regime of the Cadman, Funk and Lever Fjords, however these studies demonstrate that sills do not need to block all warm water to restrict melt rates. Done – we have modified the discussion to be richer and clearer around this point, reflect the detailed factors at play and cite additional references: 397: ‘The entrances to the Lever and Funk fjords have bathymetric sills at depths of 200 m and 230 m respectively, which limit the access of deep warm water to these glaciers compared with Cadman, so they are likely to be less sensitive to ocean forcing than Cadman Glacier which does not have a sill higher than the 350 m mid bay ridge. Sills are an effective mechanism to restrict deep, warm oceanic waters to marine-terminating glaciers, and modelling work shows that sills deeper than warm water layers can still limit basal melt rates where a hydraulically controlled circulation regime dominates. Our results suggest differences in bathymetric structure plays a strong role in the different response of Cadman and neighbouring glaciers to warming shelf water.’
3.11	311-312	What year is the ocean temperature anomaly shown?	Fig. 6b	Done - Anomaly in Fig 6 is 2019. Added year to Fig 6b legend to make this clearer
3.12	326-329	It would be helpful to include a figure reference	334	Done – added reference to Fig 4a,b
3.13	328-329	What is the postulated driver of this long term thinning? Ocean warming?	335	Done – changed to 335: ‘This, combined with previous studies in the area, suggests that the glacier experienced persistent ocean driven thinning of its ice shelf’
3.14	371-373	Again it would be helpful to include a figure reference	385	Done – added a reference to Fig. 2

References

Adusumilli, S. *et al.* (2018) ‘Variable Basal Melt Rates of Antarctic Peninsula Ice Shelves, 1994–2016’, *Geophysical Research Letters*, 45(9), pp. 4086–4095. Available at: <https://doi.org/10.1002/2017GL076652>.

Bao, W. and Moffat, C. (2023) 'Impact of shallow sills on heat transport and stratification regimes in proglacial fjords', *The Cryosphere Discussions*, pp. 1–24. Available at: <https://doi.org/10.5194/tc-2023-32>.

Brunt, K.M. *et al.* (2010) 'Mapping the grounding zone of the Ross Ice Shelf, Antarctica, using ICESat laser altimetry', *Annals of Glaciology*, 51(55), pp. 71–79. Available at: <https://doi.org/10.3189/172756410791392790>.

Fricker, H.A. and Padman, L. (2006) 'Ice shelf grounding zone structure from ICESat laser altimetry', *Geophysical Research Letters*, 33(15). Available at: <https://doi.org/10.1029/2006GL026907>.

Friedl, P. *et al.* (2020) 'Remote sensing of glacier and ice sheet grounding lines: A review', *Earth-Science Reviews*, 201, p. 102948. Available at: <https://doi.org/10.1016/j.earscirev.2019.102948>.

Gourmelen, N. *et al.* (2017) 'Channelized Melting Drives Thinning Under a Rapidly Melting Antarctic Ice Shelf', *Geophysical Research Letters*, 44(19), pp. 9796–9804. Available at: <https://doi.org/10.1002/2017GL074929>.

Hogg, A.E. *et al.* (2018) 'Mapping ice sheet grounding lines with CryoSat-2', *Advances in Space Research*, 62(6), pp. 1191–1202. Available at: <https://doi.org/10.1016/j.asr.2017.03.008>.

Howat, I.M. *et al.* (2019) 'The Reference Elevation Model of Antarctica', *The Cryosphere*, 13(2), pp. 665–674. Available at: <https://doi.org/10.5194/tc-13-665-2019>.

Jakobsson, M. *et al.* (2020) 'Ryder Glacier in northwest Greenland is shielded from warm Atlantic water by a bathymetric sill', *Communications Earth & Environment*, 1(1), pp. 1–10. Available at: <https://doi.org/10.1038/s43247-020-00043-0>.

Moffat, C. and Meredith, M. (2018) 'Shelf–ocean exchange and hydrography west of the Antarctic Peninsula: a review', *Philosophical Transactions of the Royal Society A: Mathematical, Physical and Engineering Sciences*, 376(2122), p. 20170164. Available at: <https://doi.org/10.1098/rsta.2017.0164>.

Mohajerani, Y. *et al.* (2021) 'Automatic delineation of glacier grounding lines in differential interferometric synthetic-aperture radar data using deep learning', *Scientific Reports*, 11. Available at: <https://doi.org/10.1038/s41598-021-84309-3>.

Nilsson, J. *et al.* (2022) 'Hydraulic suppression of basal glacier melt in sill fjords', *EGU sphere*, pp. 1–33. Available at: <https://doi.org/10.5194/egusphere-2022-1218>.

Schaffer, J. *et al.* (2020) 'Bathymetry constrains ocean heat supply to Greenland's largest glacier tongue', *Nature Geoscience*, 13(3), pp. 227–231. Available at: <https://doi.org/10.1038/s41561-019-0529-x>.

Turner, J. *et al.* (2022) 'Record Low Antarctic Sea Ice Cover in February 2022', *Geophysical Research Letters*, 49(12), p. e2022GL098904. Available at: <https://doi.org/10.1029/2022GL098904>.

van Wessem, J.M. *et al.* (2016) 'The modelled surface mass balance of the Antarctic Peninsula at 5.5 km horizontal resolution', *The Cryosphere*, 10(1), pp. 271–285. Available at: <https://doi.org/10.5194/tc-10-271-2016>.

REVIEWERS' COMMENTS

Reviewer #1 (Remarks to the Author):

This is a re-review.

As I noted in my initial review, this is a well-done paper that assembles several data sets for a detailed analysis of a relatively important western Antarctic Peninsula glacier.

In my view, all of the reviewer comments were well-addressed. In particular, I appreciated the discussion of Reverwer2's suggestion of Eulerian and Lagrangian melt tracking -- without very dense altimetry data (in both time and space) this would not be easily possible, nor would such a result materially change the results presented. Reviewer 3 has only a few substantive comments. All three reviewers expressed appreciation of the work.

Reviewer 3 pointed out that this paper might not be apropos for Nature Communications, and there is some merit to that -- Nature Geoscience might have been a better choice,, with essentially the same overall impact on the physical science community. However, similar works have appeared in Nature Communications in the past, and are highly cited and I think appreciated by the wider earth and climate research community.

Reviewer #2 (Remarks to the Author):

The Authors have addressed all my comments, I encourage the editor to accept this paper for publication.

Reviewer #3 (Remarks to the Author):

Thank you to the authors for carefully and clearly responding to the comments I raised on the previous version of this manuscript. These were largely relatively minor queries and suggestions, which the authors have addressed satisfactorily. I am also pleased to see some increased discussion of the wider significance of the findings.

I have only a few comments on this revised manuscript, along with a number of suggestions relating to the clarity of the writing, which I will attach as an annotated PDF.

Comments:

L266-7. The discharge here is described as being constant between 2016 and 2018, when Figure 5 shows it varies quite substantially in this time period. I think what is meant is that there is no trend of

increasing ice discharge over this time period – if so, the phrasing should be amended accordingly.

L363-5. This indicates that one would assume that surface ponding was a necessary condition for ice shelf collapse, but this seems to be a bit of an Antarctic Peninsula–centric perspective. The laterally constrained ice shelf at Cadman may have more in common with the Greenland’s relatively small ice shelves and tongues, where surface melt ponds are not generally deemed to have such an important part to play, a comparison that should probably be made here.

L396-425. A missing link here seems to be to make the connection between the divergent behaviour of Cadman, Lever and Funk Glaciers and the discussion around tipping points. Yes Lever and Funk Glaciers may be less sensitive to ocean conditions than Cadman due to the shallower sills, but this may be only part of the story. As discussed in this section, the rapid changes at Cadman Glacier may reflect the culmination of a lengthy period of mass imbalance in the terminal region, which is finally sufficient to trigger a dramatic dynamic response. Lever and Funk may also be slowly responding to ocean warming, but not presently in a position whereby this is sufficient to trigger a major response. A comparison can be made to Cadman Glacier around 1995 – if you only had the data for that period in Figure 2, you might argue that Cadman Glacier was insensitive to ocean warming, but then an ocean warming event of similar magnitude 25 years later caused the tongue to collapse. Similarly, Lever and Funk may seem insensitive now to ocean warming now, but this could dramatically change at some point down the line. It would be interesting to see if there were any more gradual changes in ice thickness apparent on the tongues of these glaciers if they fall within the area covered by your data. At the least though, I think this is a point that merits mention in the Discussion.

Figure 4. The new markers in part a are better, but they are still the same colour as the topographic shading. Why not use a different colour, like green or pink?

Response to second round of reviewers' comments: Ocean warming drives rapid dynamic activation of a marine-terminating glacier on the west Antarctic Peninsula

Ref: NCOMMS-23-19496A

Reviewer 1:

Ref.	Original Line	Comment	Revised Line	Response
1.1	n/a	This is a re-review. As I noted in my initial review, this is a well-done paper that assembles several data sets for a detailed analysis of a relatively important western Antarctic Peninsula glacier. In my view, all of the reviewer comments were well-addressed. In particular, I appreciated the discussion of Reverwer2's suggestion of Eulerian and Lagrangian melt tracking -- without very dense altimetry data (in both time and space) this would not be easily possible, nor would such a result materially change the results presented. Reviewer 3 has only a few substantive comments. All three reviewers expressed appreciation of the work. Reviewer 3 pointed out that this paper might not be apropos for Nature Communications, and there is some merit to that -- Nature Geoscience might have been a better choice, with essentially the same overall impact on the physical science community. However, similar works have appeared in Nature Communications in the past, and are highly cited and I think appreciated by the wider earth and climate research community.	N/A	We thank the reviewer for their time reviewing this manuscript and for their kind and constructive comments. No further changes are required to address these comments.

Reviewer 2:

Ref.	Original Line	Comment	Revised Line	Response
2.1	N/A	The Authors have addressed all my comments, I encourage the editor to accept this paper for publication.	N/A	We thank the reviewer for their time reviewing this paper and their helpful comments. No further changes are required to address these comments.

Reviewer 3:

Ref.	Original Line	Comment	Revised Line	Response
3.1	N/A	Thank you to the authors for carefully and clearly responding to the	N/A	

		comments I raised on the previous version of this manuscript. These were largely relatively minor queries and suggestions, which the authors have addressed satisfactorily. I am also pleased to see some increased discussion of the wider significance of the findings. I have only a few comments on this revised manuscript, along with a number of suggestions relating to the clarity of the writing, which I will attach as an annotated PDF.		
3.2	266-267	L266-7. The discharge here is described as being constant between 2016 and 2018, when Figure 5 shows it varies quite substantially in this time period. I think what is meant is that there is no trend of increasing ice discharge over this time period – if so, the phrasing should be amended accordingly.	266	Thank you for this comment, we agree that changing the phrasing to be more precise would be better. Changed to: ‘Throughout 2015 ice discharge increased until it reached a mean of 1.85 ± 0.13 Gt/yr between 2016 and 2018 with no positive or negative trend’
3.3	363-365	L363-5. This indicates that one would assume that surface ponding was a necessary condition for ice shelf collapse, but this seems to be a bit of an Antarctic Peninsula-centric perspective. The laterally constrained ice shelf at Cadman may have more in common with the Greenland’s relatively small ice shelves and tongues, where surface melt ponds are not generally deemed to have such an important part to play, a comparison that should probably be made here.	371	Thank you, we agree that the precise meaning could be made clearer. Changed to: 370: ‘The absence of surface melt ponding on the Cadman Ice Shelf prior to its collapse demonstrates that surface ponding is not a necessary precondition for ice shelf collapse in the Antarctic Peninsula, which can occur through ocean-driven thinning and unpinning alone, similar to small ice shelf failures observed in Greenland’
3.4	396-425	L396-425. A missing link here seems to be to make the connection between the divergent behaviour of Cadman, Lever and Funk Glaciers and the discussion around tipping points. Yes Lever and Funk Glaciers may be less sensitive to ocean conditions than Cadman due to the shallower sills, but this may be only part of the story. As discussed in this section, the rapid changes at Cadman Glacier may reflect the culmination of a lengthy period of mass imbalance in the terminal region, which is finally sufficient to trigger a dramatic dynamic response. Lever and Funk may also be slowly responding to ocean warming, but not presently in a position whereby this is sufficient to trigger a major response. A comparison can be made to Cadman Glacier around 1995 – if you only had the data for that period in Figure 2, you might argue that Cadman Glacier was insensitive to ocean warming, but then an ocean warming event of similar magnitude 25 years later caused the tongue to collapse. Similarly, Lever and Funk may seem insensitive now to ocean warming now, but this could dramatically		Changed to: 410: ‘Our results suggest differences in bathymetric structure played a strong role in the different response of Cadman and neighbouring glaciers to warming shelf water, however in future Lever and Funk Glaciers may become substantially more sensitive to ocean forcing if temperature anomalies are greater and they progressively thin in an evolution which is similar to Cadman Glacier’s but delayed.’

		change at some point down the line. It would be interesting to see if there were any more gradual changes in ice thickness apparent on the tongues of these glaciers if they fall within the area covered by your data. At the least though, I think this is a point that merits mention in the Discussion.		
3.5		Figure 4. The new markers in part a are better, but they are still the same colour as the topographic shading. Why not use a different colour, like green or pink?		Done: Figure 4a markers changed to gold with black outline and made larger
3.6	16	suggest 'the rapid acceleration and collapse'		We chose to use 'increase in speed' because this period of net acceleration stopped after December 2019 and features some brief periods of slowing speed superimposed on the positive trend. Given there is not a constant ongoing acceleration, we have chosen to use 'increase in speed'.
3.7	19	Of?		We have chosen to keep the original phrasing.
3.8	19	By?		We have chosen to keep the original phrasing.
3.9	20	Awkward sentence		We have chosen to keep the original phrasing.
3.10	23	Unclear terminology – imbalance of what?	22	Done: Added 'dynamic imbalance'
3.11	23	grammar - 'increased ice discharge from glaciers on the Antarctic Peninsula' would be preferable.	23	Done: Changed to the suggested wording.
3.12	27	Awkward (and unnecessary) to start and finish with reference to time frames - suggest just one or the other	27	Done: changed to: 'Research on the Antarctic Peninsula (AP) has shown major changes in the region's glaciers and ice shelves throughout the 20th and 21st centuries.'
3.13	31	Off		We have chosen to keep the original phrasing.
3.13	40	Predicted		Unchanged, as we have already said projected: 'and ice loss is projected to increase in the future...'
3.14	44	Complex sentence		We have chosen to keep the original phrasing.
3.15	54	Delete 'the'	54	Done
3.16	78	Delete 'these are'	78	Done
3.17	124	'ice speed' – unnecessary?	124	Done
3.18	203	'ice speed' – unnecessary?		Maintained here for clarity as we refer to surface elevation change which could also be accelerating.
3.19	212	Comma	212	Added
3.20	219	Check grammar and punctuation for clarity		
3.21	260	Change 'on' to 'from'	264	Done
3.22	260	Add: 'the'		We have chosen to keep the original phrasing.
3.23	264	Meaning unclear – do you mean by the time of some observation that spans this time period?		We feel this sentence is adequately clear